# A recombinant protein containing influenza viral conserved epitopes and superantigen induces broad-spectrum protection

**Yansheng Li**[1,2,3], **Mingkai Xu**[1,3]*, **Yongqiang Li**[1,2,3], **Wu Gu**[1,3], **Gulinare Halimu**[1,2,3], **Yuqi Li**[1,2,3], **Zhichun Zhang**[1,2,3], **Libao Zhou**[4], **Hui Liao**[4], **Songyuan Yao**[4], **Huiwen Zhang**[1,3], **Chenggang Zhang**[1,3]

[1]Institute of Applied Ecology, Chinese Academy of Sciences, Shenyang, China; [2]University of Chinese Academy of Sciences, Beijing, China; [3]Key Laboratory of Superantigen Research, Shenyang Bureau of Science and Technology, Shenyang, China; [4]Chengda Biotechnology Co. Ltd, Liaoning, China

**Abstract** Influenza pandemics pose public health threats annually for lacking vaccine that provides cross-protection against novel and emerging influenza viruses. Combining conserved antigens that induce cross-protective antibody responses with epitopes that activate cross-protective T cell responses might be an attractive strategy for developing a universal vaccine. In this study, we constructed a recombinant protein named NMHC that consists of influenza viral conserved epitopes and a superantigen fragment. NMHC promoted the maturation of bone marrow-derived dendritic cells and induced CD4+ T cells to differentiate into Th1, Th2, and Th17 subtypes. Mice vaccinated with NMHC produced high levels of immunoglobulins that cross-bound to HA fragments from six influenza virus subtypes with high antibody titers. Anti-NMHC serum showed potent hemagglutinin inhibition effects to highly divergent group 1 (H1 subtype) and group 2 (H3 subtype) influenza virus strains. Furthermore, purified anti-NMHC antibodies bound to multiple HAs with high affinities. NMHC vaccination effectively protected mice from infection and lung damage when exposed to two subtypes of H1N1 influenza virus. Moreover, NMHC vaccination elicited CD4+ and CD8+ T cell responses that cleared the virus from infected tissues and prevented virus spread. In conclusion, this study provides proof of concept that NMHC vaccination triggers B and T cell immune responses against multiple influenza virus infections. Therefore, NMHC might be a candidate universal broad-spectrum vaccine for the prevention and treatment of multiple influenza viruses.

*For correspondence: mkxu@iae.ac.cn

## Editor's evaluation

The authors provided a new concept for making a universal influenza vaccine by fusing T cell epitopes, B cell epitopes, and superantigen. They conclude that their construct is effective against broad influenza strains, and now the convincing evidence for their conclusion has been provided.

## Introduction

Seasonal influenza viruses infect 5–15% of the global population and kill several hundred thousand people every year despite the availability of antivirals and inactivated tetravalent vaccines, which are effective for most recipients (*Ellebedy et al., 2014*). As a cluster of RNA viruses, influenza viruses have a segmented genome and error-prone RNA-dependent RNA polymerase, which enables

influenza viruses to undergo minor antigenic changes (antigenic drift) and major antigenic changes (antigenic shift) (*Wang et al., 2010*). This genomic mutability of influenza viruses permits the virus to evade adaptive immune responses. The unpredictable variability of influenza A viruses causes annual epidemics in human populations, and the time-consuming methods used to develop and produce influenza vaccines have resulted in a lack of effective prevention against influenza infection (*Lu et al., 2014*). Moreover, currently available vaccines induce antibodies against most homologous virus strains, but do not protect against antibody-escape variants of seasonal or novel influenza viruses. Therefore, the development of a novel universal vaccine that induces broad protection against various influenza viruses is urgently required to overcome the problems caused by the annual epidemic of different types of influenza viruses. Conserved proteins or fragments of influenza A virus such as nucleoprotein (NP), matrix protein 2 ectodomain (M2e), and hemagglutinin stem (HA2) (*Staneková and Varečková, 2010*; *El Bakkouri et al., 2011*; *De Filette et al., 2006*), which induce cross-protective immune responses, were reported to be promising antigens for the design of a universal vaccine.

Hemagglutinin is a highly immunogenic surface protein in the influenza virus. Overall, 18 HA subtypes of influenza A viruses have been identified and can be classified into two groups (12 subtypes from group 1 including H1 and H5, and 6 subtypes from group 2 including H3 and H7) based on the phylogenetic relationships of the HA proteins. HA consists of two subunits, a membrane-distal globular domain HA1 and a conserved stalk domain HA2 linked by a single disulfide bond (*Wilson et al., 1981*). The epitopes of broadly neutralizing antibodies (bnAbs) in HA2 are more conserved across different influenza HA subtypes compared with the antigenic sites in HA1 (*Julien et al., 2012*; *Ellebedy and Ahmed, 2012*). The bnAbs induced by HA2 recognize diverse influenza A virus subtypes and prevent fusion of the virus and host cell membranes (*Julien et al., 2012*). M2, a membrane protein, forms a pH-gated proton channel incorporated into the viral lipid envelope, which is essential for the efficient release of the viral genome during virus entry (*Schnell and Chou, 2008*). M2e, a 23 amino acid peptide, is highly conserved in influenza A strains and is thought to be a valid and versatile vaccine candidate that can induce heterosubtypic antibody responses against various human influenza strains. NP, a conserved inner antigen of the influenza virus, is required to induce cellular immune responses after natural infection. Furthermore, NP-specific helper T cells augment protective antibody responses and promote B cells to produce HA-specific antibodies (*Townsend and Skehel, 1984*). Therefore, an effective vaccine against seasonal or pandemic influenza should contain conserved B-cell epitopes that prime the humoral immune system to induce a response that significantly diminishes virus replication immediately after infection, and T cell epitopes that promote cellular immune responses for overall protection (*Moltedo et al., 2009*; *Hermesh et al., 2010*).

Bacterial superantigen staphylococcal enterotoxin C2 (SEC2) can directly activate T lymphocytes without the presence of antigen-presenting cells (APCs) and stimulate bone marrow-derived dendritic cells (BMDCs) maturation (*Yao et al., 2018b*; *Dinges et al., 2000*). Thus, SEC2 can activate T cells and BMDCs to produce various cytokines such as interleukin-4 (IL-4) and interferon-gamma (IFN-γ). However, very low amounts of SEC2 can induce CD8+ cytotoxic T lymphocytes (CTLs) to specifically kill target cells such as virus-infected cells and tumor cells (*Fu et al., 2020*; *Laidlaw et al., 2013*). Furthermore, our previous study reported that SEC2 linked innate immunity and adaptive immunity by activating Toll-like receptor (TLR) downstream signaling molecules including MyD88 and NF-κB, and might be a promising adjuvant for rabies vaccines to provide efficient protection against exposure to the lethal rabies virus. Taken together, we hypothesized that SEC2 has adjuvant or adjuvant-like effects that would improve the protective efficiency of influenza vaccines (*Yao et al., 2018a*).

We designed a universal influenza vaccine by connecting the conserved fragments of NP (two epitopes of CTL and helper T lymphocytes), M2e, and highly conserved long α-helix regions of HA2 from group 1 (H1) and group 2 (H3 and H7) using a flexible linker (GSAGSAG) to construct a fusion protein NMH as a subunit vaccine (*Figure 1A*). To enhance the antigenicity of NMH, we fused SEC2 into the C-terminal of NMH to construct a conjugate vaccine NMHC. We evaluated the influenza virus-specific antibody-inducing ability of NMHC using a direct-binding ELISA. Microneutralization-hemagglutination inhibition and cytotoxic lysis assays were performed to evaluate the bnAbs and immunogens induced by NMHC. We also evaluated the universal protective efficiency of NMHC in mice infected with influenza viruses in vivo.

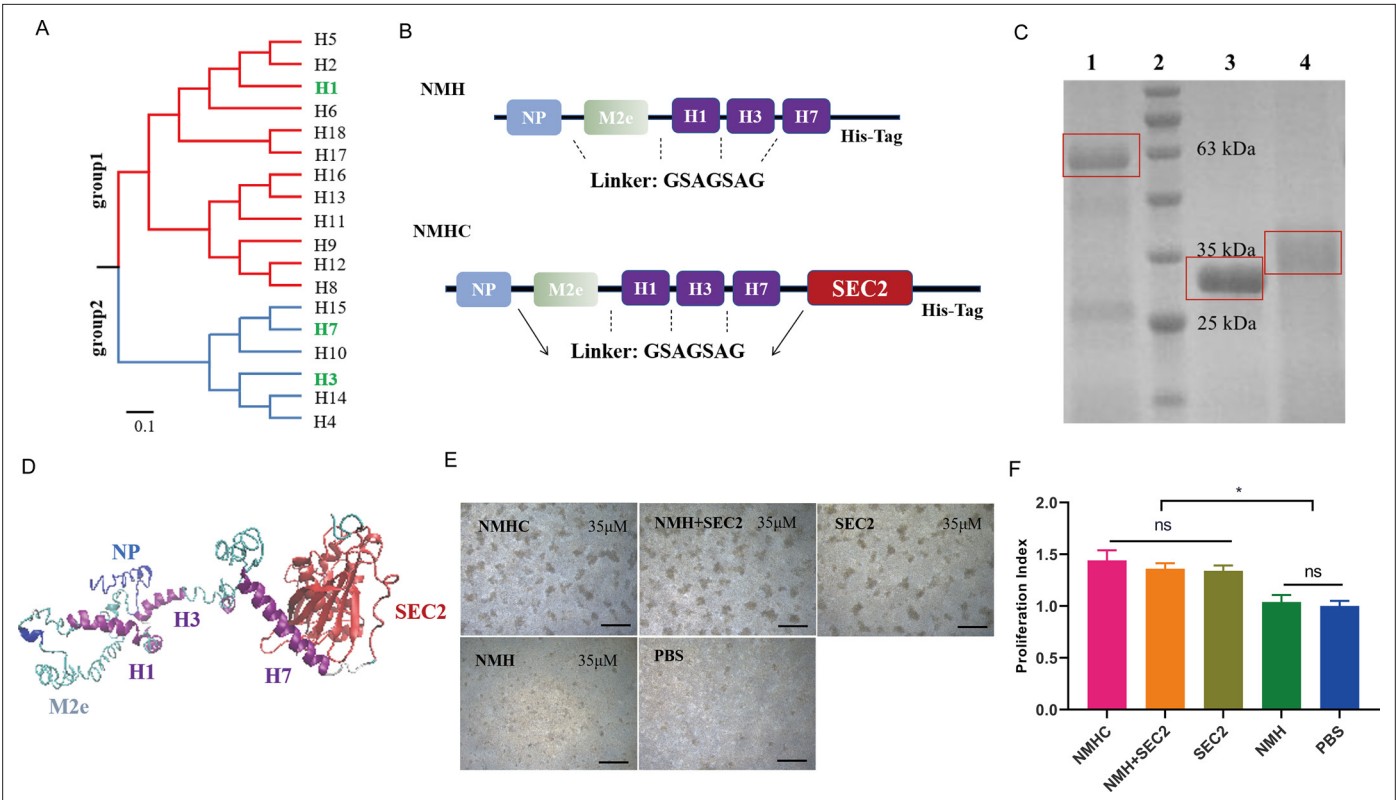

**Figure 1.** Design, purification, renaturation, and verification of the recombinant protein NMHC. (**A**) Phylogenetic tree of the 18 hemagglutinin subtypes of influenza A viruses based on amino acid sequences. Group 1 and group 2 subtypes are listed in red and blue, respectively, and the HAs used in recombinant protein NMHC are listed in green. The amino acid distance scale bar denotes a distance of 0.1. (**B**) NMH consists of the conserved gene segments of NP, M2e, and HA2, which are connected by linker (GSAGSAG). NMHC consists of NMH and linker followed by SEC2. (**C**) NMHC, SEC2, and NMH purified and renatured from BL21(DE3) lysate. Lane 1: renatured NMHC; lane 2: marker; lane 3: purified SEC2; lane 4: renatured NMH. (**D**) The structure of the recombinant protein NMHC about homology modeling. The light blue, light green, and purple portions are NP, M2e, and HA2(H1, H3, H7) fragments, respectively, and red portion fragment is SEC2 domain. (**E**) Murine splenocytes proliferation induced by NMHC, NMH + SEC2, SEC2, NMH, and PBS were observed at 72 hr. (**F**) Proliferation index was quantified by MTS assay. Scale bar = 200 μm. Data are represented as mean ± SD (n = 3). *$p < 0.05$, ns, not significant. Source files of the gel used for the qualitative analyses are available in *Figure 1—source data 1*.

The online version of this article includes the following figure supplement(s) for figure 1:

**Source data 1.** Source file for the gel data used for the qualitative analyses of NMHC, SEC2, and NMH purified and renatured from BL21(DE3) lysate shown in *Figure 1*.

## Results

### Construction, production, and renaturation of recombinant proteins

The recombinant protein NMHC consisted of NMH at the N-terminal and SEC2 at the C-terminal, fused with a flexible linker sequence. To facilitate purification, a carboxyl-terminal His-tag reading frame was retained on the backbone of the expression vector pET-28a (+) (*Figure 1B*). Recombinant proteins were expressed as inclusion bodies in plasmid-harboring *Escherichia coli* strains and purified as a major band at 59 kDa (NMHC) and 31 kDa (NMH) by Coomassie blue-stained SDS-PAGE (*Figure 1C*). After renaturation by dialysis, the soluble recombinant proteins NMHC and NMH were obtained with >95% purity confirmed by SDS-PAGE. The structural modeling of NMHC in silico revealed that the fused NMH was independent of the SEC2 region (*Figure 1D*), which implied that the domains of NMH and SEC2 would not affect each other. To verify this hypothesis, a splenocyte proliferation assay was performed. As shown in *Figure 1E*, NMH did not induce proliferation compared with the phosphate buffer saline (PBS)-negative control, whereas SEC2, NMHC, and NMH + SEC2 induced obvious proliferative clusters of murine splenocytes at 72 hr. The MTS result demonstrated that the proliferation index (PI) of the NMHC group was significantly higher than that of the NMH and

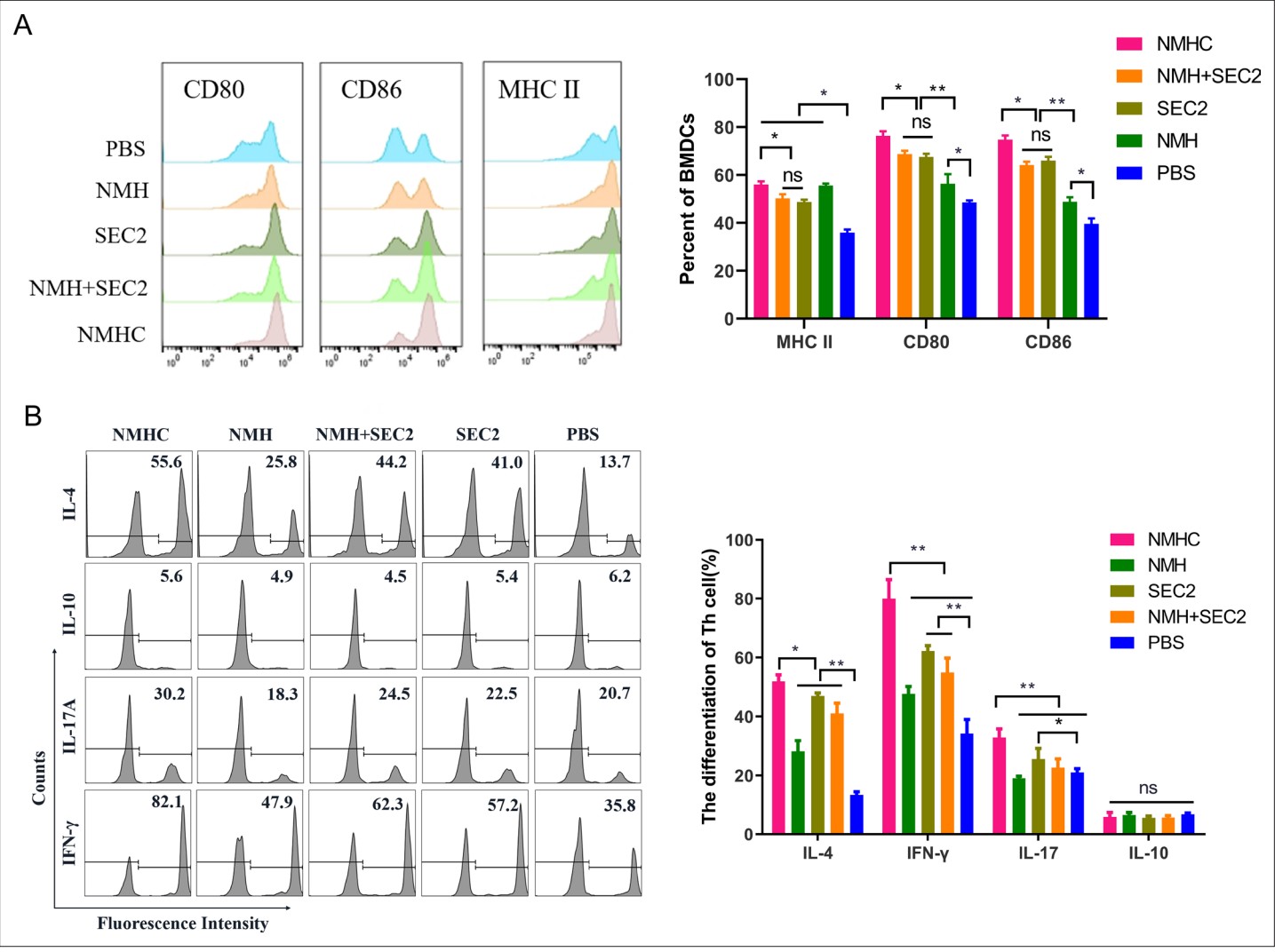

**Figure 2.** The expressions of co-stimulatory molecules CD80, CD86, and MHC II induced by recombinant proteins on BMDCs that inducted CD4⁺ T cell differentiation in vitro. (**A**) BMDCs were treated with NMHC, NMH + SEC2, SEC2, NMH, and PBS as control for 48 hr. Then BMDCs were stained with antibodies to MHC II, CD80, CD86, and analyzed with flow cytometry. (**B**) Recombinant proteins-treated BMDCs (10⁴) were co-cultured with CD4⁺ T cells (10⁵) for 96 hr, and CD4⁺ T cells differentiations were analyzed with flow cytometry. The results were showed with gated percentage. Data are represented as mean ± SD (n = 3). *p<0.05, **p<0.01, ns, not significant.

PBS groups (p<0.05, *Figure 1F*) and that there was no significant difference in these terms between the SEC2, NMH + SEC2, and NMHC groups.

## The biological activity of recombinant proteins in vitro

BMDCs are potent APCs that connect the innate and adaptive immune responses. The maturation of murine BMDCs was evaluated by detecting surface markers including CD80, CD86, and MHC II. CD4⁺ T cell differentiation mediated by matured BMDCs was evaluated by flow cytometry. As shown in *Figure 2A*, the expressions of CD80, CD86, and MHC II in the NMHC, NMH + SEC2, SEC2, and NMH groups were significantly increased compared with the PBS control group, which implied that BMDC maturation was induced by these treatments. BMDCs treated with NMHC had significantly increased expressions of CD80, CD86, and MHC II compared with the NMH + SEC2 and SEC2 groups (p<0.05). Furthermore, the expressions of CD80 and CD86 in the NMHC and NMH + SEC2 groups were significantly higher than those in the NMH group (p<0.01), although no statistical difference was noted between the SEC2 and NMH + SEC2 groups (p>0.05). As shown in *Figure 2B*, mature BMDCs induced by NMHC, SEC2, NMH + SEC2, and NMH significantly promoted CD4⁺ T cells to express

the Th1 cytokine IFN-γ and Th2 cytokine IL-4 compared with the PBS control group (p<0.01), and the NMHC group had higher expressions of IFN-γ, IL-4, and the Th17 cytokine IL-17 compared with the SEC2, NMH + SEC2, and NMH groups (p<0.05 or p<0.01). No statistical difference was noted in the expression of the regulatory T cell cytokine IL-10 between all groups (p>0.05). This result implied that mature BMDCs induced by NMHC promoted CD4$^+$ T cells to differentiate into Th1, Th2, or Th17 cell subtypes but not Treg cells.

## Murine serum immunoglobulin isotyping elicited by recombinant proteins

Female BALB/c mice were immunized by a prime-boost-boost schedule with 2 weeks intervals. Serum samples were collected on days 14, 28, 42, and 100, and seven isotypes of κ immunoglobulins (accounting for 95% of immunoglobulin subtypes in mice; *Pricop et al., 1994*) were detected by CBA. As shown in *Figure 3*, at 100 days post first immunization, IgG2aκ underwent significant changes. There was no statistical change in IgG2aκ in the first 14 days. On day 28, the NMHC, NMH, and NMH + SEC2 groups exhibited significantly enhanced production of IgG2aκ compared with the SEC2 and PBS groups (p<0.05 for NMHC and NMH, and p<0.01 for NMH + SEC2), and the NMH + SEC2 treatment group had the highest level of IgG2aκ. Similar results were observed on day 42, but the highest level of IgG2aκ was in the NMHC group. Over the long term, the production of IgG2aκ returned to normal in all groups at day 100, although the NMHC group exhibited significantly enhanced IgG2aκ production compared with the PBS control group. There was no difference in IgG2aκ between the SEC2 and PBS groups at all time points tested, which suggested that IgG2aκ detected in this study was specifically induced by NMH antigen but not by SEC2. These data demonstrated that the vaccination induced immunogen-boosting IgG2a, which indicated T cell-dependent antibody production and affinity matured antivirus responses (*Wang et al., 2010*).

## Recombinant proteins induced neutralizing antibody in mice

Because stem-directed bnAbs mediated neutralization by inhibiting membrane fusion but not by blocking the virus from binding to receptors on the host cells (*Julien et al., 2012*), the standard hemagglutinin inhibition assay is not fit for this study (as in *Supplementary file 1a*, neutralizing antibody was not detected in serum at the lowest dilution of 1:8). To evaluate the titers of neutralizing antibody induced by recombinant proteins, an optimized microneutralization-hemagglutinin inhibition assay was performed as described in Methods. As shown in *Figure 4*, NMHC, NMH + SEC2, and NMH induced high levels of neutralizing antibody titers after the third immunization, which prevented replication of the Bris/02(H1), Kan/14(H3), MI/45(H1), and HK/4801(H3) viruses compared with SEC2 and PBS. Furthermore, NMHC and NMH + SEC2 exhibited higher titers than NMH. This result suggested that sera from recombinant protein immunized mice had substantial heterosubtypic neutralizing activity, and that the mechanism of protection might be related to inhibiting virus-host cell membrane fusion (*Okuno et al., 1994*).

## Recombinant proteins induced binding antibodies to a broad panel of HA subtypes

To determine whether immunization with recombinant proteins induced binding antibodies to a broad panel of HA subtypes, ELISAs were performed with four recombinant HAs and two split virions. As expected, NMHC, NMH + SEC2, and NMH induced high levels of antibodies against various HA subtypes, whereas SEC2 and PBS did not induce any effective antibodies (*Figure 5A–F*, *Supplementary file 1b*). NMHC and NMH + SEC2 had greater potency at inducing H3-, H5-, and H7-specific antibodies compared with NMH (*Figure 5C–E*), and NMHC had a higher induction ability for H1-specific antibodies compared with NMH + SEC2 and NMH (*Figure 5A*). Moreover, taking the MI/45(H1) and HK/4801(H3) strains as an example, we also measured influenza-specific IgG1 and IgG2a subclasses in antisera, which are essential for understanding B cell somatic hypermutation and subclass switching (*Quan et al., 2007*). The results showed that NMHC induced higher H1N1-specific IgG1 and IgG2a levels than NMH and NMH + SEC2 (*Figure 5G and H*), and that NMHC and NMH + SEC2 induced higher H3N2-specific IgG1 and IgG2a levels than NMH (*Figure 5I and J*), which was consistent with the results of the total IgG (*Figure 5A and C*). These data indicate that NMHC significantly induced broad binding or neutralizing antibody activity and might confer cross-subtype protection.

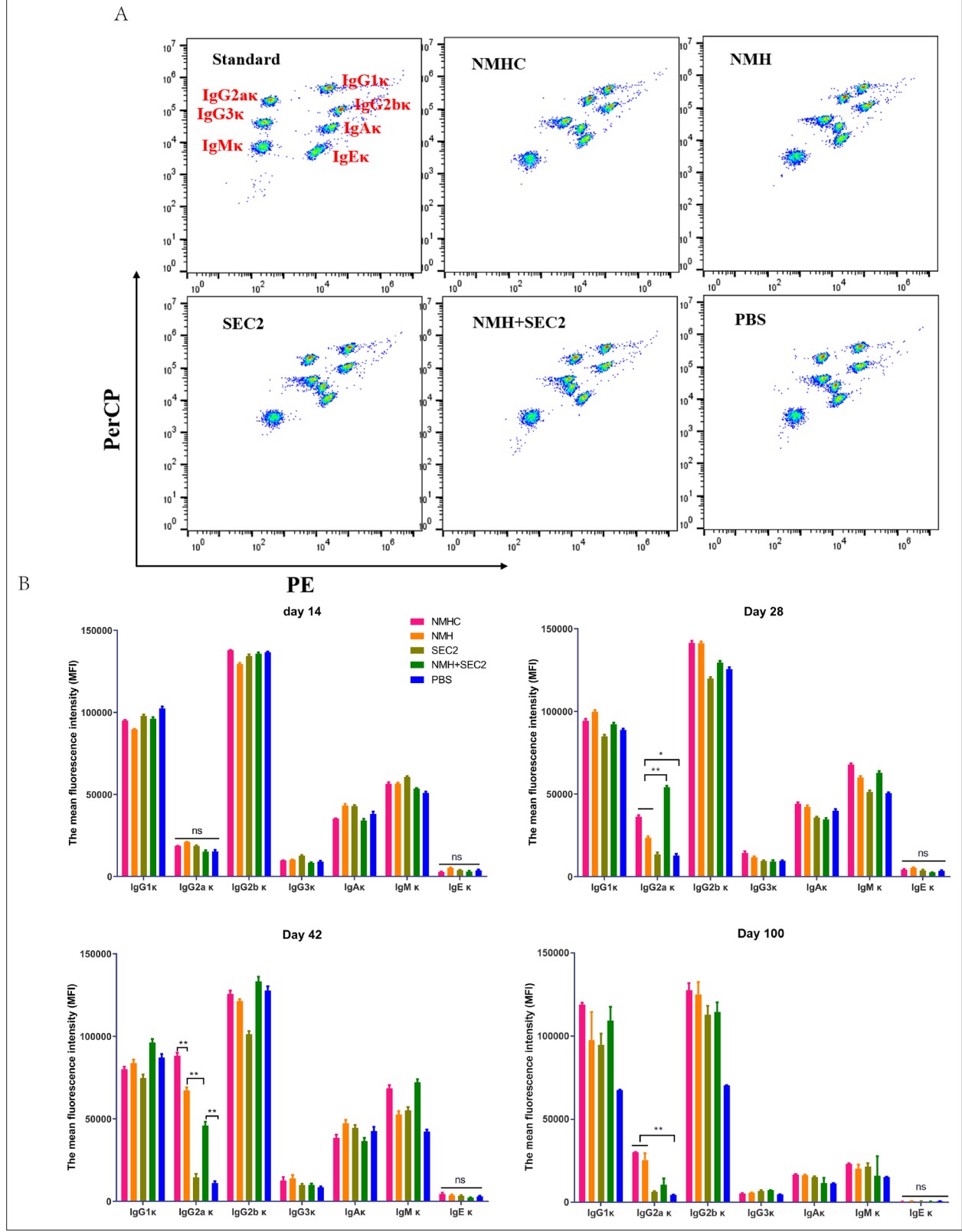

**Figure 3.** Murine serum immunoglobulin isotyping elicited by recombinant proteins. Sera were taken from immunized mice on days 14, 28, 42, and 100 after immunization, and immunoglobulins were examined by Cytometric Bead Array. (**A**) Seven clusters of beads represented the immunoglobulins of IgG1 κ , IgG2a κ , IgG2b κ , IgG3 κ , IgA κ , IgM κ , and IgE κ  from top to bottom when detected in FL2 (PE) and FL3 channels (PerCP). (**B**) The contents of immunoglobulins in different time points were represented by median fluorescence intensity. Data are represented as mean ± SD (n = 3). *p<0.05, **p<0.01, ns, not significant.

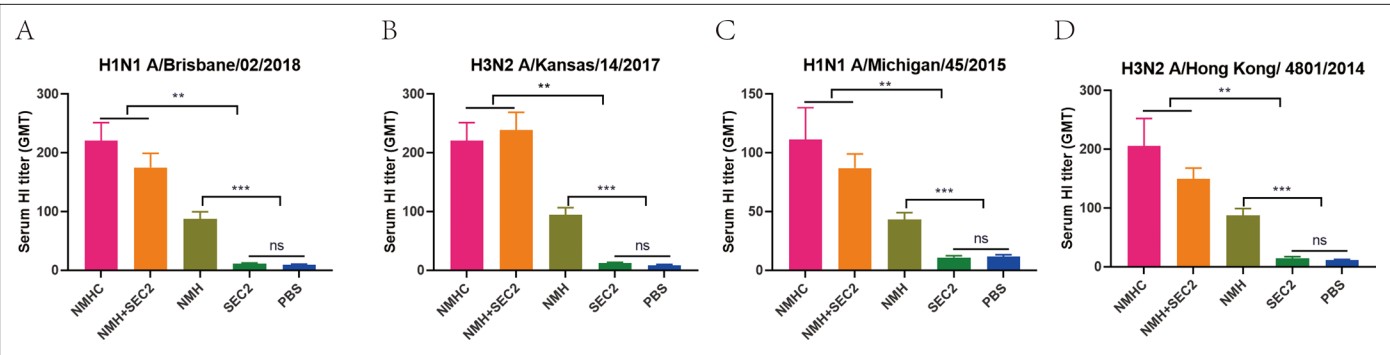

**Figure 4.** Recombinant proteins elicit broadly cross-reactive bnAbs in mice. Sera were taken from immunized mice on day 42 after immunization, and the neutralization assays were performed against Bris/02(H1) (**A**), Kan/14(H3) (**B**), MI/45(H1) (**C**), and HK/4801(H3) (**D**) influenza viruses. The titers of each serum sample were defined as the reciprocal of the highest dilution. GMT, geometric mean titer. *p<0.05, **p<0.01, ***p<0.001, ns, not significant.

**Figure 5.** Breadth of the antibody response elicited by the recombinant protein was determined by ELISA of the pooled antisera against purified rHA proteins or split virion. (**A–F**) Total IgG against (**A**) H1N1 A/Michigan/45/2015, (**B**) H2N2 A/Canada/720/2005, (**C**) H3N2 A/Hong Kong/4801/2014, (**D**) H5N1 A/Hubei/1/2010, (**E**) H7N9 A/Shanghai/2/2013, (**F**) H9N7 A/Hong Kong/35820/2009. (**G–J**) IgG1 or IgG2a specific against split virion of influenza viruses. (**G**) IgG1 specific against H1N1 A/Michigan/45/2015, (**H**) IgG2a specific against H1N1 A/Michigan/45/2015, (**I**) IgG1 specific against H3N2 A/Hong Kong/4801/2014, and (**J**) IgG2a specific against H3N2 A/Hong Kong/4801/2014.

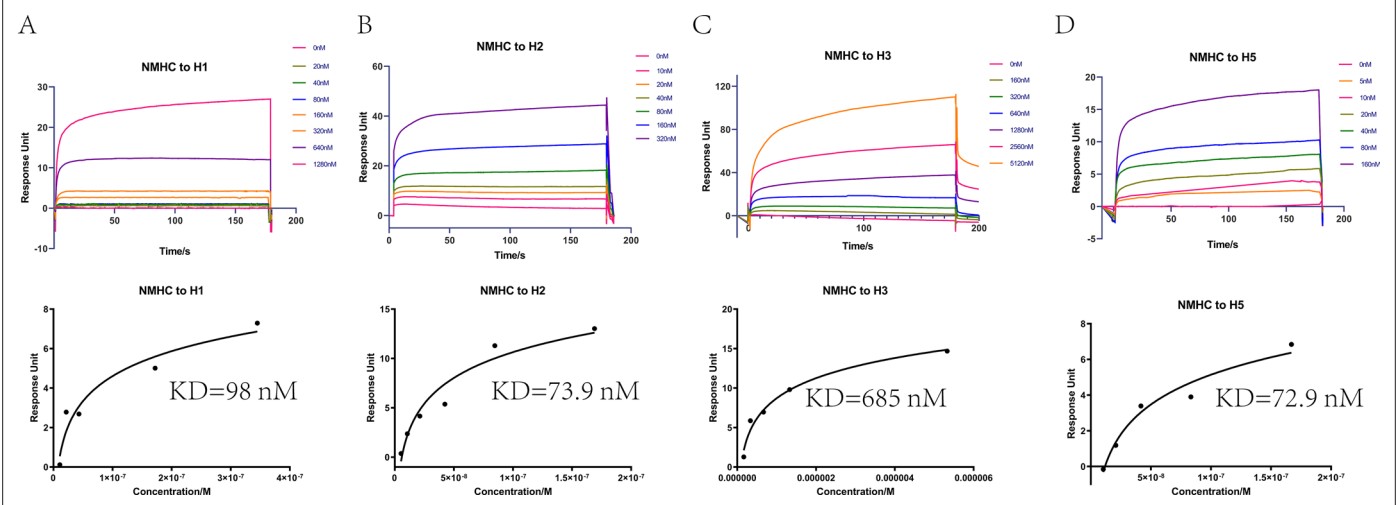

**Figure 6.** Affinities of HAs to antibodies purified from anti-NMHC sera. The overlays of binding kinetics of the HAs at different concentrations and the affinity of KD values determined with Biacore were shown. (**A**) Split virion of H1N1 A/Michigan/45/2015, (**B**) H2N2 A/Canada/720/2005, (**C**) split virion of H3N2 A/Hong Kong/4801/2014, and (**D**) H5N1 A/Hubei/1/2010. The KD values were calculated using a steady affinity state model by the Biacore T200 evaluation software (version 3.1). Source files of the overlays of binding kinetics used for the affinity analyses are available in *Figure 6—source data 1*.

The online version of this article includes the following figure supplement(s) for figure 6:

**Source data 1.** Source file for affinities data shown in *Figure 6*.

**Figure supplement 1.** NMH forms a stable complex with the purified antibodies from serum of immunized mice.

**Figure supplement 1—source data 1.** Source file for pull-down assay shown in *Figure 6—figure supplement 1*.

## NMHC induced antibodies with high binding affinity to various HAs

We purified the serum antibodies and evaluated their ability to form a stable complex with the NMH antigen. The results of pull-down assays showed that antibodies purified from NMHC, NMH + SEC2, and NMH antisera, but not from SEC2 and PBS antisera, bound to and pulled down NMH antigen (*Figure 6—figure supplement 1*). Subsequently, SPR experiments were performed to quantify the binding affinities of antibodies purified from anti-NMHC sera. NMHC-induced Abs displayed potent binding affinity to H1, H2, H3, and H5 fragments with KD values of 98, 73.9, 685, and 72.9 nM, respectively (*Figure 6*). This result suggests that immunization with NMHC elicited broadly cross-reactive protection against various influenza viruses.

## Recombinant proteins induced ASC responses

ELISpot assays were performed to measure antigen-specific B-cell responses induced by recombinant proteins (*Figure 7A*). Of note, H1N1 and H3N2 viruses were components of the seasonal vaccines for the 1978–2020 influenza seasons. First, we determined the kinetics of H1N1- and H3N2-specific ASCs in pre- and post-immunized mice (*Figure 7B and C*). The responses indicated a positive correlation with vaccination time and peaked at day 42. At the peak of the response, the frequencies of NMH-, H1N1-, and H3N2-specific ASCs were significantly higher in the NMHC, NMH + SEC2, and NMH immunized groups than in the SEC2 and PBS-treated groups (p<0.01, Figure 7D–F), and there was no statistical difference between the SEC2 and PBS groups (p>0.05). Moreover, at the peak, the frequencies of NMH-, H1N1-, and H3N2-specific ASCs induced by NMHC were 73 ± 5.2, 48 ± 5.7, and 64 ± 2.1 per million, respectively, which were approximately 1.5-fold higher than the frequencies of those induced by NMH (p<0.05, Figure 7D–F). Of note, NHMC immunization induced higher frequencies of H1N1- and H3N2-specific ASCs compared with NMH + SEC2 immunization (p<0.05, *Figure 7E and F*). Interestingly, in comparison with the prime vaccination, we observed a large increase in the frequency of anti-H1N1 and H3N2 ASCs at day 100 in the NMHC-immunized group (*Figure 7B and C*), suggesting that NMHC induces cross-reactive memory B cells after the third immunization (*Purtha et al., 2011*). In summary, NMHC effectively induced antigen-specific B-cell responses and elicited serological memory that might be maintained by long-lived antibody-secreting plasma cells and

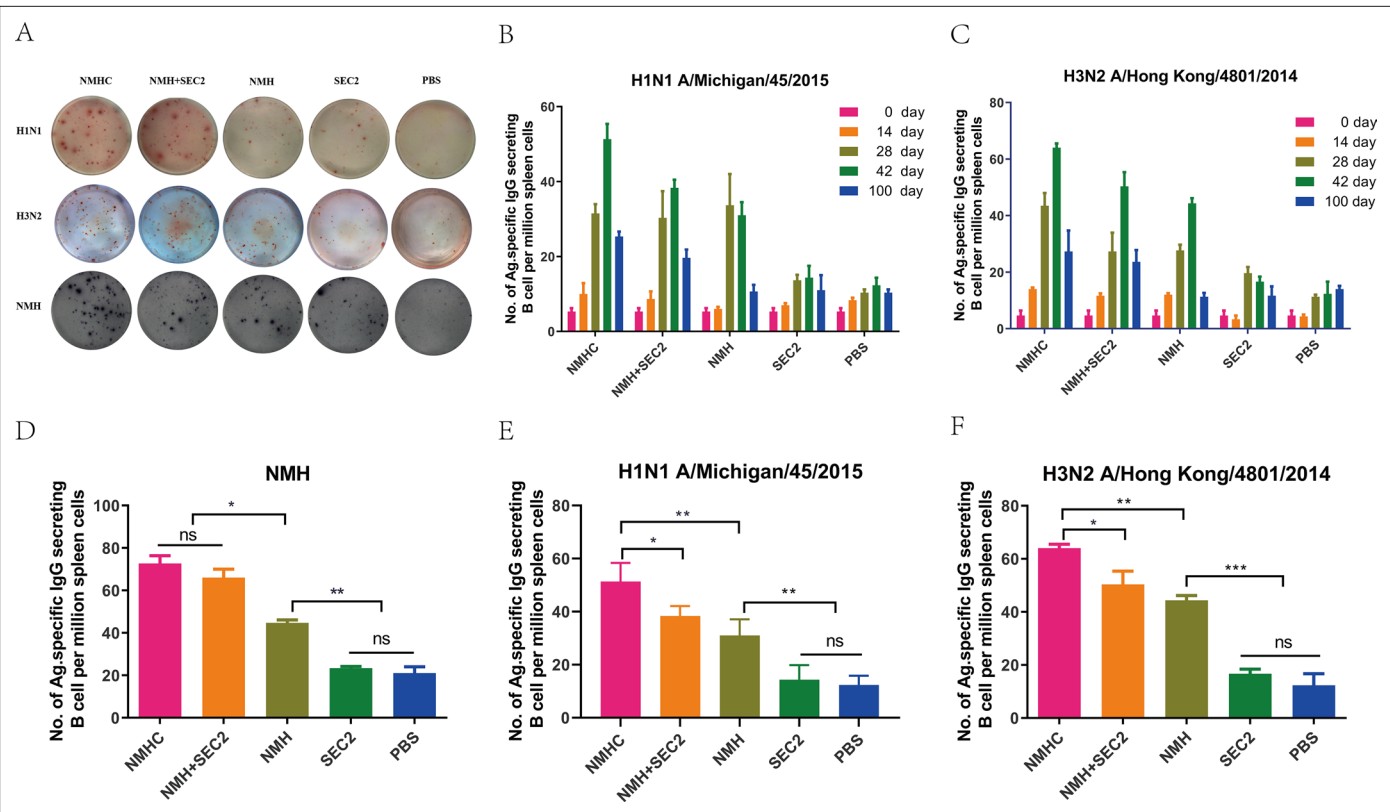

**Figure 7.** The specific antibody-secreting cells' (ASC) response to H1N1, H3N2, and NMH was measured with ELISpot assay. Splenocytes were isolated and collected from immunized mice on days 14, 28, 42, and 100. The cells were added in ELISpot wells, which coated with the NHM proteins, H1N1 or H3N2 influenza viruses. (**A**) Each spot in the well represents an ASC. (**B**, **C**) The frequency of H1N1 and H3N2-specific ASC on days 0, 14, 28, 42, and 100. (**D–F**) Specific ASCs' response to NMH, H1N1, and H3N2 influenza viruses the frequency of NMH-, H1N1-, and H3N2-specific ASC on day 42. Data are represented as mean ± SD (n = 3). *p<0.05, **p<0.01, ***p<0.001, ns, not significant.

reinforced by memory B cells, which can rapidly differentiate into antibody-secreting cells (ASCs) when re-exposed to the specific antigen.

## Recombinant proteins induced T cell responses

We studied the vaccine-induced $CD4^+$ and $CD8^+$ T cell immune responses, which are potent mediators of heterosubtypic immunity against different influenza viruses. Because IFN-γ and IL-4 are typically produced by Th1 and Th2 cells, respectively (*Yao et al., 2018a*), ELISpot assays were performed to measure IFN-γ or IL-4-secreting cells induced by recombinant protein immunization. Immunization with NMHC and NMH + SEC2 significantly induced the upregulation of IFN-γ or IL-4-producing splenocytes (p<0.01 compared with the PBS group; *Figure 8A*), and there was no statistical difference compared with the SEC2 group (p>0.05). Furthermore, immunization with NMH alone did not induce any cytokine-producing splenocytes compared with the PBS group (p>0.05). In addition, IFN-γ and IL-4 as well as the Th1- and Th2-specific transcription factors T-bet and GATA3, respectively, were quantified by qPCR. The results were consistent with those from the ELISpot assay where NMHC, NMH + SEC2, and SEC2 significantly induced the upregulation of IFN-γ and IL-4 transcription compared with NMH alone and PBS (p<0.05). Similar changes were found for the T-bet and GATA3 transcription factors. To verify the antigen-specific T cell responses, we detected IL-4 and IFN-γ cytokine production after stimulating splenocytes from post-immunized mice with 1000 ng/mL NMH or 1000 ng/mL SEC2. Splenocytes from the NMHC, NMH + SEC2, and NMH immunized groups treated with NMH produced significantly higher levels of IL-4 and IFN-γ than with CK treatment (p<0.05). No significant changes were observed in the SEC2 and PBS immunized groups. This result indicated that the T cell responses induced by NMH treatment were influenza antigen-specific. However, splenocytes from all immunized groups produced high levels of IL-4 and IFN-γ after SEC2 treatment in vitro, and there

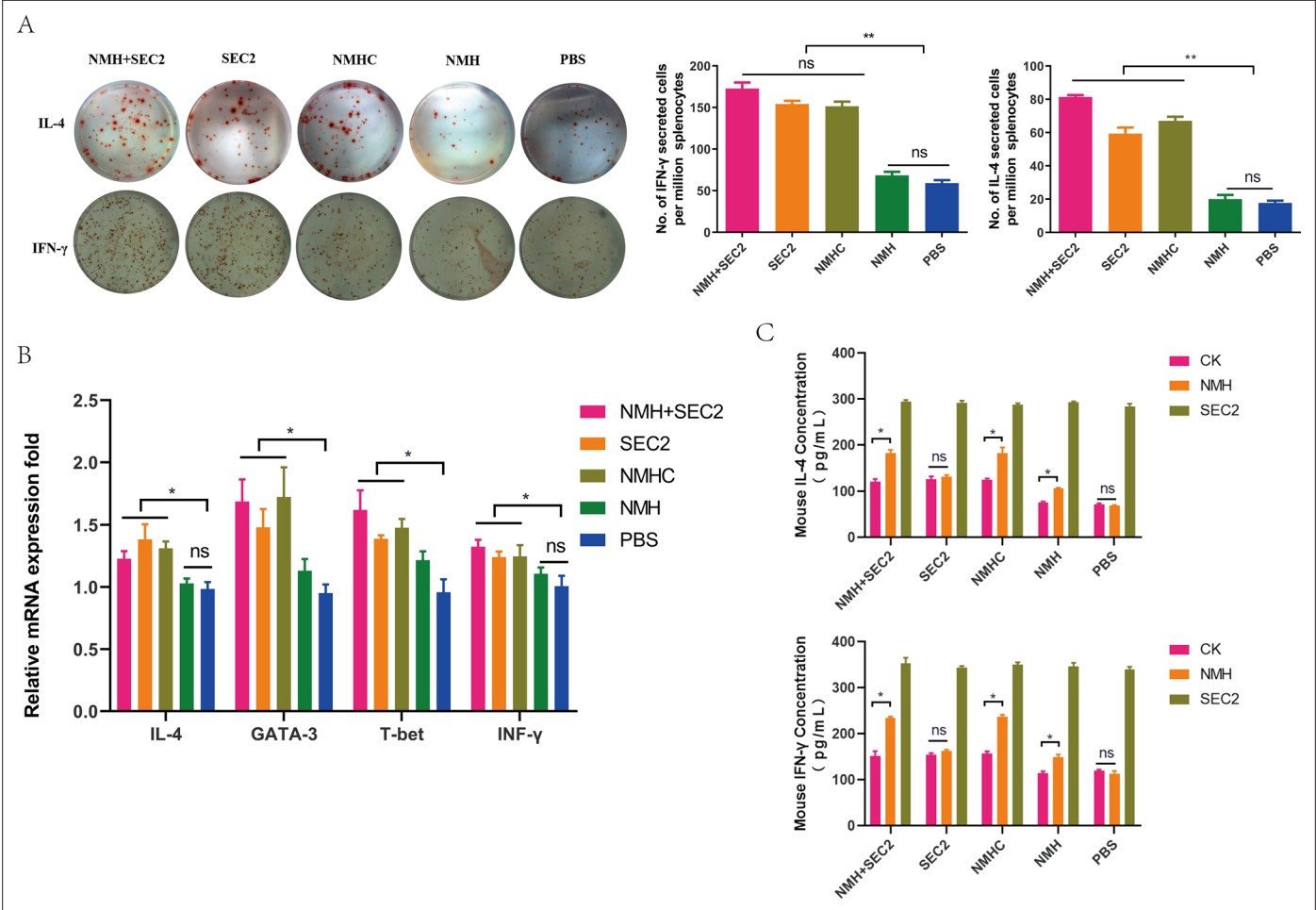

**Figure 8.** Recombinant proteins induced CD4[+] T cell responses. (**A**) Splenocytes isolated from immunized mice in each group on days 42 were plated in ELISpot wells to detected IFN-$\gamma$ and IL-4 cytokine-producing cells. (**B**) The mRNA levels of IFN-$\gamma$, IL-4, T-bet, and GATA3 were measured by qPCR. (**C**) IL-4 and IFN-$\gamma$ produced by splenocytes treated with NMH and SEC2 respectively in vitro were detected by ELISA. PBS was served as CK. Results are expressed as the mean value ± SD (n = 3). *p<0.05, **p<0.01, ns, not significant.

were no significant differences between the five immunized groups. As a superantigen, SEC2 induced nonspecific T cell responses, leading to the massive production of cytokines. Taken together, NMHC immunization activated Th1 and Th2 cells, which are necessary for immune responses to clear virus infection.

The cytotoxic effects of CD8[+] T cells are important for clearing virus-infected cells, and thus have important roles in the efficient control of influenza virus spread (*Yao et al., 2018a*). To assess whether the recombinant vaccines could augment antigen-specific CD8[+] T cell responses, equal numbers of CFSE[high] target cells and CFSE[low] non-target control cells were mixed and injected intravenously into recipient mice for 12 hr and specific CTL were examined in vivo. First, we noted that Alexa Fluor 488-labeled NMH antigens were internalized into splenocytes isolated from naïve mice detected by LSCM (*Figure 9A*). Second, target cells and control cells were easily distinguished when labeled with 10 μM or 1 μM of CFSE, and the fluorescence intensity of the former was higher than the latter (*Figure 9B*). These results indicate that the relative number of residual target cells (CFSE[high]) isolated from recipient mice immunized with NMHC and NMH + SEC2 was markedly reduced compared with those immunized with NMH, SEC2, or PBS (p<0.01; *Figure 9C*). Furthermore, NMHC significantly induced perforin and granzyme expression at the transcription level, which also confirmed that NMHC induced strong antigen-specific cytotoxic responses (*Figure 9D*).

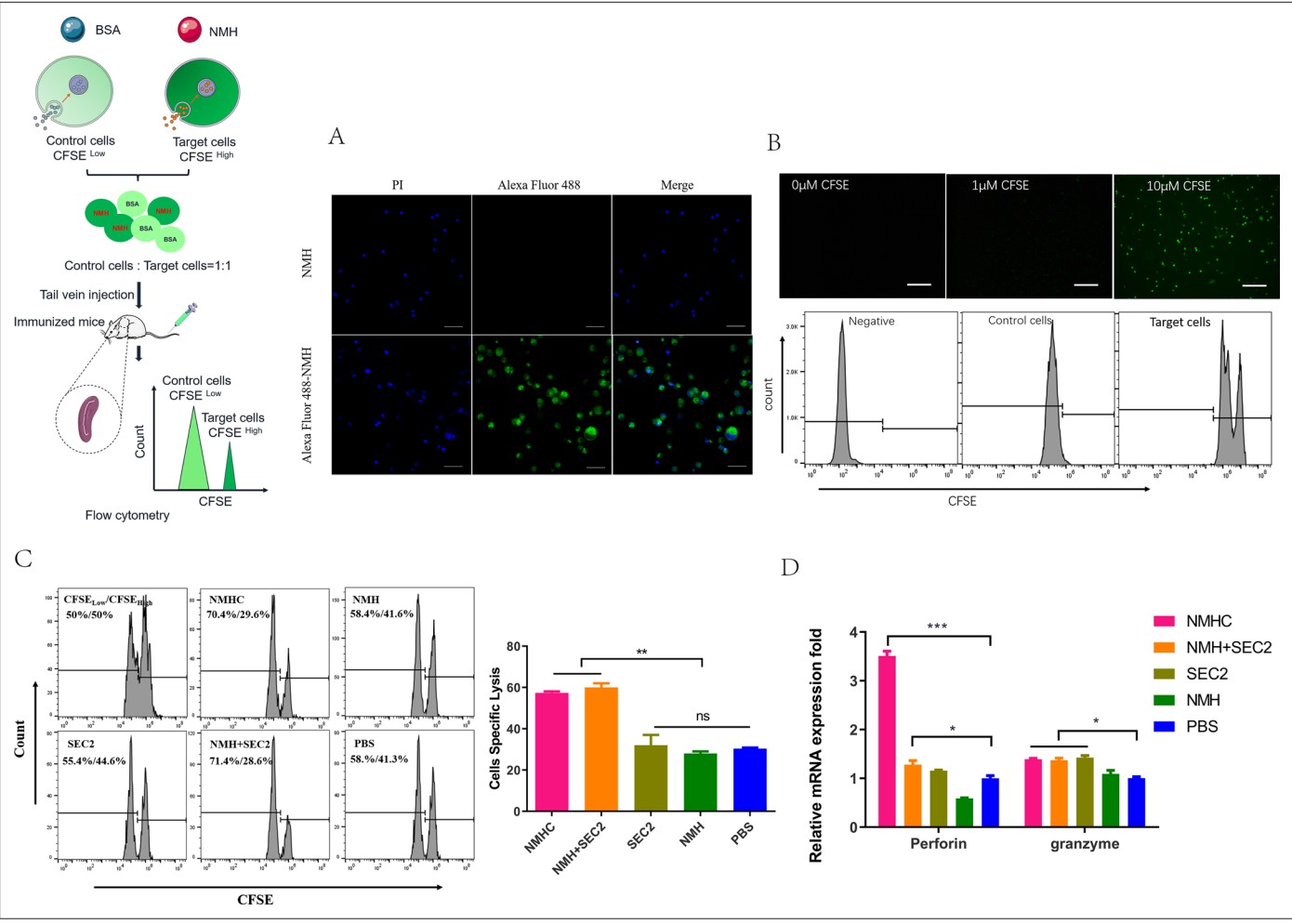

**Figure 9.** Effects of the recombinant protein NMHC on CD8[+] T cell responses. (**A**) To verify that NMH antigens can be effectively internalized into splenocytes, splenocytes from naïve BALB/c mice were co-cultured with Alexa Fluor 488-labeled NMH or NMH alone and detected by fluorescent inverted microscope. (**B**) Then, fresh isolated naïve splenocytes were divided into two parts. One part was pulsed with 5 μg/mL NMH, labeled with 10 μM of CFSE and termed as CFSE high target cells, the other part was loaded with BSA, labeled with 1 μM of CFSE and termed as CFSE low control cells, both of them detected by fluorescent inverted microscope and flow cytometry. (**C**) The equal number of two parts cells was mixed and injected into recipient mice. 12 hr later, the splenocytes from these mice were collected to examine the antigen-specific cytolytic responses. (**D**) The mRNA levels of perforin and granzyme of recipient mice were examined by qPCR. Statistical analyses were performed using Student's t-test and one-way ANOVA. *p<0.05, **p<0.01, ***p<0.001, ns, not significant.

## Immunized with recombinant proteins provide in vivo protection against influenza virus challenge

To evaluate whether the recombinant protein immunization conferred cross-strain protection, mice were challenged with virus strains of Bris/02(H1) or MI/45(H1) 2 weeks after the third immunization. At 6 days post-infection, the qPCR analysis of the lung tissues demonstrated that the mice immunized with NMHC exhibited up to a 4 log reduction in viral HA gene copy number compared with the PBS-treated groups (*Figure 10A and B*). Mice immunized with NMH + SEC2 showed similar results. During the challenge experiment, only PBS-treated groups had a reversible loss of body weight (*Figure 10C and D*). Histopathological analysis demonstrated no visible pathological changes in the lungs of NMHC-immunized mice at 6 days post-infection, whereas the PBS control group exhibited severe alveolar damage and interstitial inflammatory infiltration (*Figure 10E and F*).

We examined the lymphocyte infiltration in the lungs of mice by immunohistochemistry (IHC) to evaluate the inflammatory damage caused by influenza virus infection. As expected, the leukocyte common antigen CD45 was widely detected in the lung sections of PBS-treated mice but not in naïve mice without infection (p<0.05 for Bris/02(H1) and p<0.01 for MI/45(H1), *Figure 11*). Immunization

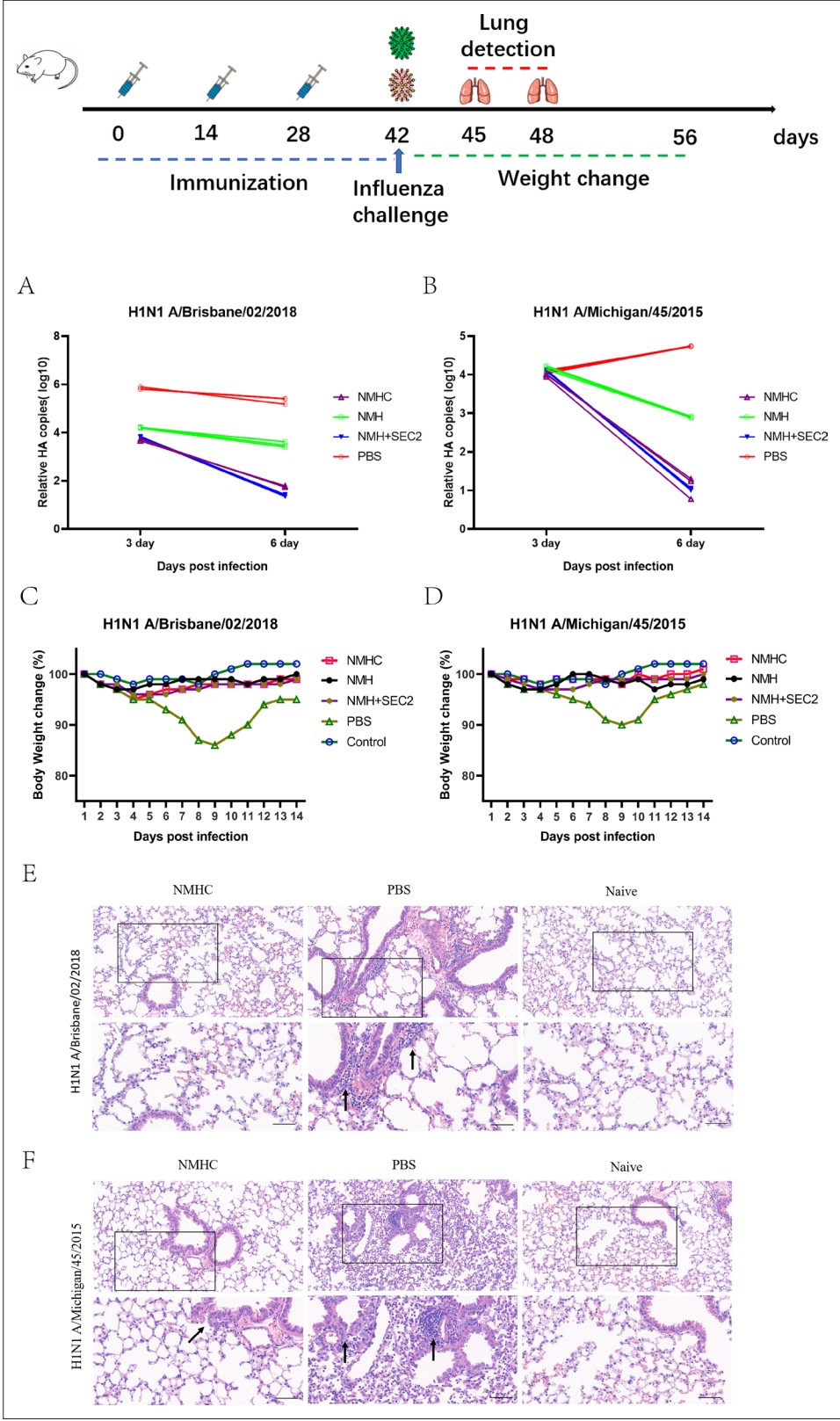

**Figure 10.** Challenge test in BALB/c mice. Mice (n = 12 per group in two separate experiments) were infected intranasally with $10^3$ TCID$_{50}$ of Bris/02(H1) or MI/45(H1) influenza viruses after 14 days of the third Immunization. (**A**, **B**) Viral copy number of Bris/02(H1) and MI/45(H1) in the lungs at days 3 and 6 post-infection was determined using qPCR. (**C**, **D**) Weight changes (%) were monitored for 14 days post-challenge. (**E**, **F**) Histopathology in pulmonary tissue (scale bar = 50 μm). Naïve mice that were untreated and unchallenged were used as the control.

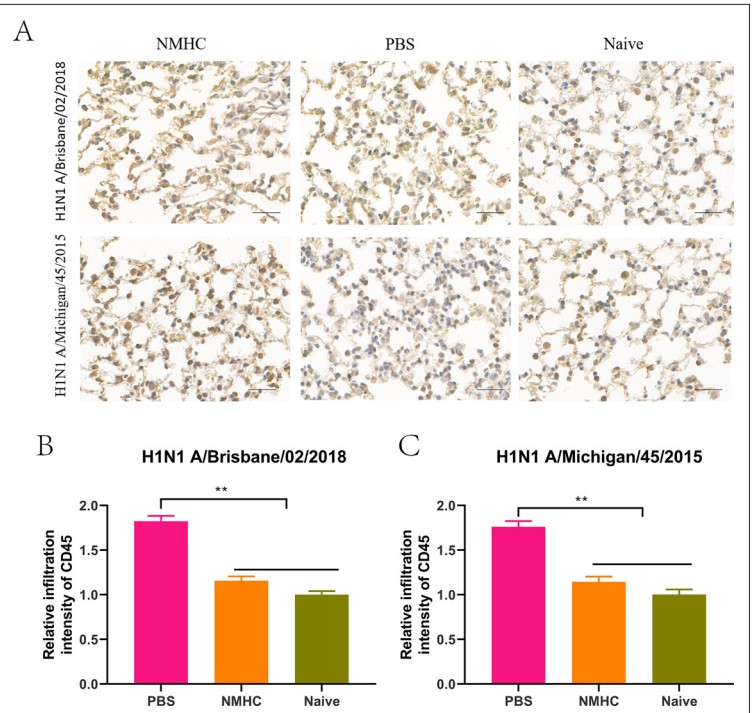

**Figure 11.** The profile analysis of lung-infiltrating leukocytes in lung frozen sections of infected mice treated with NMHC. (**A**) Lung-infiltrating leukocytes were analyzed by IHC of CD45. (**B**, **C**) The quantification of the relative infiltration intensity of biomarkers by normalizing the positive signals of the treated groups to those of the naïve group. Error bars denote mean ± SD. **p<0.01. Scale bar = 50 μm.

with NMHC before influenza virus infection significantly decreased the intensities of lung-infiltrated CD45 leukocytes compared with PBS treatment (*Figure 11B and C*), indicating a reduction in inflammatory damage in the lungs.

Furthermore, immunofluorescence analysis of lung sections with fluorescence Abs against the influenza nucleoprotein, macrophage common antigen (F4/80), and CD8 was performed to evaluate the distribution of influenza viruses, lung-infiltrating macrophages, and lung- infiltrating CD8[+] CTLs (*Figure 12A*). As shown in *Figure 12B and C*, on day 6 after infection, strong pink fluorescence signals representing influenza nucleoprotein were detected in the lungs of PBS-treated mice, which indicated the extensive distribution of influenza viruses. In addition, the relative fluorescence intensities of influenza nucleoprotein in the lungs of NMHC vaccinated mice were 4.5-fold (Bris/02(H1)) and 2.5-fold (MI/45(H1)) weaker than those in the PBS-treated mice. In both infection models, NMHC immunization induced a significantly enhanced distribution of CD8[+] CTLs in the lungs compared with the PBS control groups (p<0.01), which was consistent with the results of the CTL assay. After influenza virus infection, high levels of CD8[+] CTLs in the lungs led to the effective clearance of virus-infected cells, which contributed to the low virus loads indicated by the nucleoprotein fluorescence signals. In contrast, the intensities of macrophagocyte staining were significantly higher in PBS-treated mice than in naïve and NMHC-immunized mice (p<0.05), which might be attributable to the increased inflammation and delayed viral clearance in PBS-treated mice (*Valkenburg et al., 2014*). This result demonstrated that immunization with NMHC provided robust protection against heterologous virus challenge in vivo.

## Discussion

In this study, we constructed a recombination protein, NMHC, as a pilot study to develop a novel universal influenza vaccine. NMHC consists of two domains, an NMH domain as a universal immunogen and an SEC2 domain as an adjuvant-like molecular chaperone, fused by a flexible peptide linker. Both domains do not require post-translational modification; therefore, NMHC can be conveniently

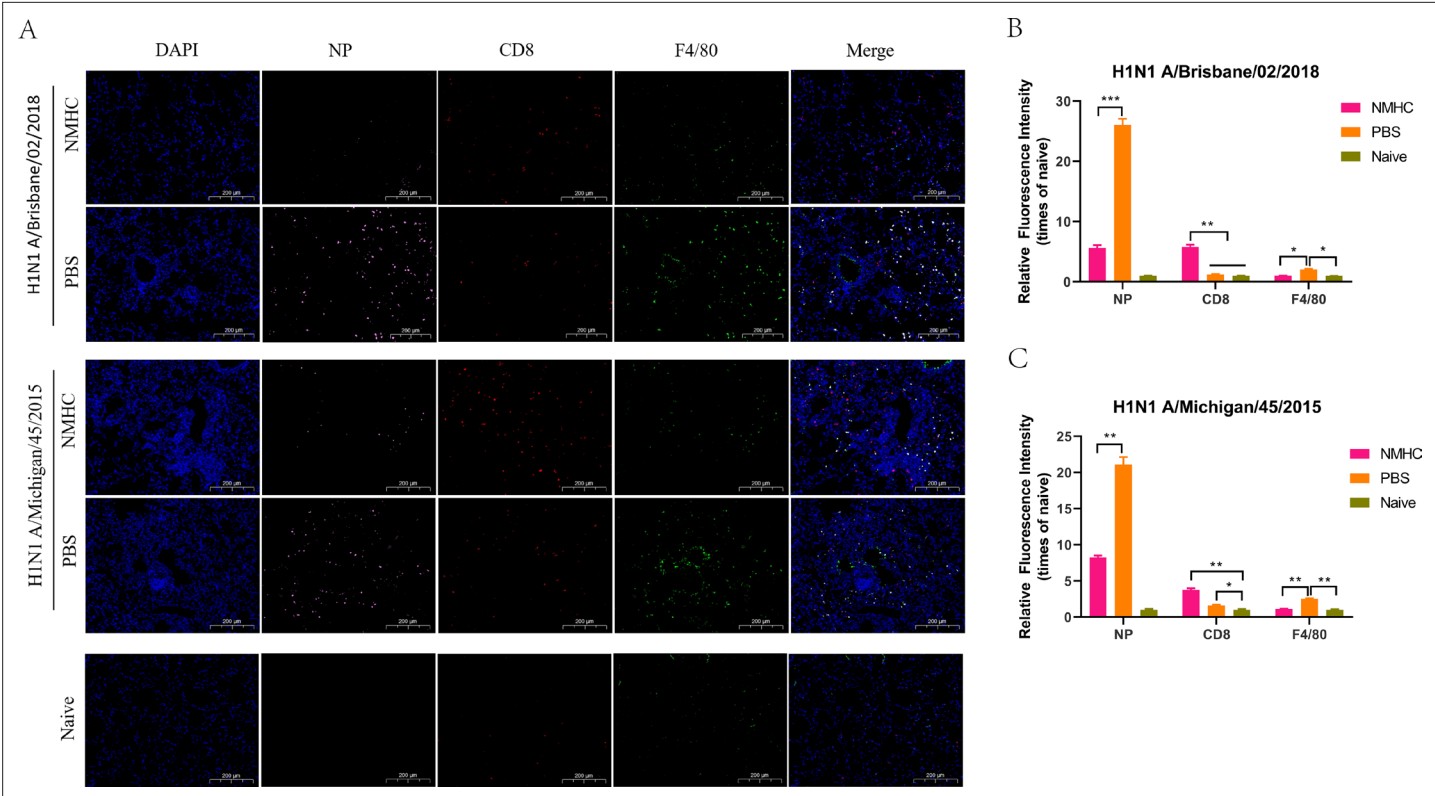

**Figure 12.** Immunofluorescence analysis of lungs frozen sections at day 6 after infection. (**A**) Distribution of influenza viruses (pink for NP), lung-infiltrating CD8[+] T cells (red for CD8), and lung-infiltrating macrophages (green for F4/80). (**B**, **C**) The relative quantification of fluorescence intensity of NP, CD8, and F4/80. The relative fluorescence intensities were quantified by normalizing the fluorescent signals of the treated groups to those of the naïve group. Error bars denote mean ± SD. Scale bar = 200 µm, *p<0.05, **p<0.01, ***p<0.001.

produced using an *E. coli* expression system. Here, we studied humoral immune responses mediated by NMHC that were involved in anti-influenza immune defense together with the contribution of cellular immunity. Our in vivo study showed that NMHC promoted B cells to differentiate into NMH-, H1N1-, and H3N2-specific ASCs or memory cells involved in humoral immune responses. NMHC induced the secretion of IgG2a and IgM, but not IgE, as the number of immunizations increased, and the antibody concentration reached a peak after the third immunization. During Th1-type immune responses, IgG2a mediates the clearance of virus and provides protection against influenza infection (*Valkenburg et al., 2014*). IgM is involved in the initial humoral immune response and IgE is associated with type I hypersensitivity. As expected, antiserum from NMHC-immunized mice specifically neutralized H1N1 and H3N2, preventing the viruses from replicating in MDCK cells (*Figure 4*). In this experiment, virus suppression by antiserum was only detected when using the MN-HI assay, but not the HI assay. A reason for this might be that stem-specific antibodies did not bind to the head region of hemagglutinin, the receptor binding site (RBS) of the influenza virus that targets cells, and therefore, did not prevent red blood cell aggregation induced by the virus (*He et al., 2015*; *van der Lubbe et al., 2018*). However, stem-specific antibodies could bind to the non-receptor binding region of HA to prevent influenza virus from fusing into the endosomal membrane by interfering with the conformational change of HA induced by a low pH, even if influenza virus had attached to the receptor and infected cells via receptor-mediated endocytosis (*Figure 13*; *Imai et al., 1998*). This procedure prevented the release of the ribonucleoprotein complex into the target cell and led to the inhibition of viral replication, so the small number of viruses could not induce red blood cell aggregation, which is directly detected by the HI assay. Next, to determine whether the antibody induced by NMHC had broad heterosubtypic binding or neutralization activity against diverse influenza A strains, we detected the binding ability of anti-NMHC sera to different HAs from group 1 (H1, H2, H5, H9) and group 2 (H3, H7) by ELISA. As expected, antiserum from NMHC-immunized mice showed a broad pattern of binding to these HA fragments with high antibody titers. Although only six types of

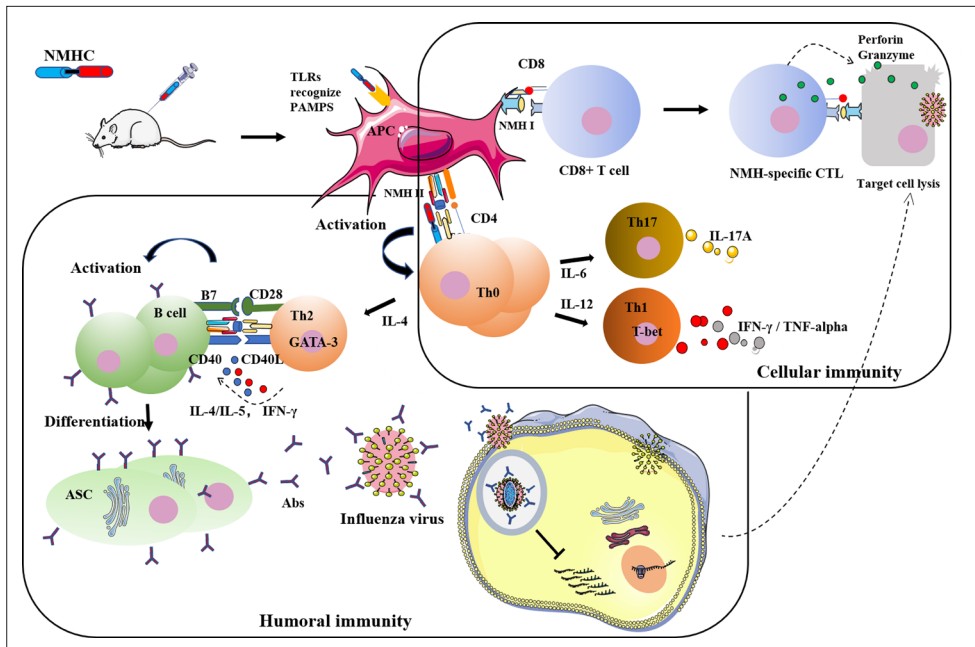

**Figure 13.** Detailed illustration of the anti-influenza mechanisms induced by NMHC.

HA were available in our experiment, we believed that the anti-NMHC serum might bind to other HA subtypes. Furthermore, the SPR assay indicated that antibodies purified from anti-NMHC serum had high binding affinities to the HAs of H1, H2, and H5, and moderate affinity to H3. Because the H3 fragment used in this study was from a split virion of H3N2 but not a commercially purified full-length fragment, we think that this relatively low affinity of antibodies to H3 might be related to the impurity of the H3 fragment. These results indicated that the antibody induced by NMHC vaccination could bind to and neutralize different influenza viruses with a high affinity and effectively prevented virus replication in target cells.

Regarding cellular immune responses, we found that NMHC promoted T lymphocyte proliferation and enhanced the functions of CD4[+] and CD8[+] T cells, respectively. CD4[+] T cells can differentiate into Th1 and Th2 cells that have critical roles in immune responses including clearing virus infection and regulating immunoglobulin isotype switching (*Snapper and Mond, 1993*). Th1 cells enhance IgG2a switching by secreting IFN-γ, and Th2 cells enhance IgG1 switching by secreting IL-4. IgG2a was reported to be more effective at resisting virus infection than IgG1 (*Yendo et al., 2016*). CD8[+] T cells enhance antiviral responses by promoting the destruction of virus-infected cells through perforin and granzyme released by CTLs. Our results indicated that immunization with NMHC significantly enhanced Th1 and Th2 immune cell responses, specifically an increase in the differentiation of IFN-γ and IL-4-producing cells and antigen-specific cytokine expression. In addition, antigen-specific CTL responses were also elevated significantly, as shown by the in vivo CTL assay, where target cells 'infected' with NMH, but not control cells 'infected' with BSA, were killed in NMHC-immunized mice. In addition, the transcription of the immune effector molecules, perforin, and granzyme was also increased.

As expected, NMHC induced the proliferation of murine splenocytes, indicating that the recombinant protein NMHC maintained the superantigen activity of SEC2. Furthermore, NMHC significantly promoted the maturation of BMDCs and increased the expressions of the co-stimulatory molecules CD80 and CD86, as well as MHC II in vitro. This effect of NMHC might be attributable to the combined effect of the NMH and SEC2 domains in the fusion protein because CD80 and CD86 expressions in BMDCs induced by NMHC were significantly higher than those induced by SEC2, NMH, and NMH + SEC2. Mature BMDCs promoted CD4[+] T cells to differentiate into Th1, Th2, and Th17 subtypes as represented by IFN-γ, IL-4, and IL-17A production. Compared with NMH and NMH + SEC2, NMHC had an adjuvant-like activity associated with the fused SEC2 domain, which regulated adaptive immune responses and augmented the immunogenicity of the NMH domain. The structural modeling

of NMHC in silico demonstrated that the fused SEC2 was independent of the NMH region, which allowed immune cells to recognize components of the antigenic determinants and induce specific immune responses. Limited by laboratory biosafety, we only used two attenuated strains of H1N1 separated by different times and places, but not other virulent pathogenic strains, in the mouse challenge experiment. Immunization with NMHC protected mice against infection by either of the two virus strains as demonstrated by the decreased number of viruses and reduction of inflammatory infiltration in the lungs. Although immunization with NMH + SEC2 achieved similar results to NMHC by reducing the number of virus particles in the lungs, we think that NMHC is preferable to NMH + SEC2 when the comprehensive effects of humoral immunity and cellular immunity, as well as the convenience of future applications, are considered.

In summary (*Figure 13*), the recombinant protein NMHC promoted B and T cell immune responses against influenza virus infection. The SEC2 domain of NMHC promoted DC maturation through the TLR/NF-κB signaling pathway (*Yao et al., 2018b*), and matured DCs presented antigens of the NMH domain to CD4+ T cells via MHC II and to CD8+ T cells via MHC I. This process led to the activation, proliferation, and differentiation of antigen-specific CD4+ T cells and CD8+ T cells. Activated antigen-specific CD4+ T cells then differentiated into IL-17A-producing Th17 cells, IFN-γ-producing Th1 cells, and IL-4-producing Th2 cells. Th1 and Th2 cells help B cells secrete HA2-specific antibodies that bind to the HA2 of the influenza virus and block the fusion of virus and the endosomal membranes of target cells to prevent release of the ribonucleoprotein complex into the cytoplasm of the target cells. In addition, Th1 cells promoted the differentiation of activated antigen-specific CD8+ T cells into CTL effector cells that specifically kill virus-infected cells by the exocytosis of cytolytic granules such as perforin and granzymes. Our findings suggest an innovative potential clinical strategy for influenza immunization. We propose the recombinant protein NMHC be a candidate universal broad-spectrum vaccine for the prevention and treatment of various influenza virus strains.

# Materials and methods

**Key resources table**

| Reagent type (species) or resource | Designation | Source or reference | Identifiers | Additional information |
|---|---|---|---|---|
| Strain, strain background (*Escherichia coli*) | BL21(DE3) | Abcam | Cat: 69450-3 | Competent cells |
| Strain, strain background (influenza virus) | H1N1 A/Michigan/45/2015 | Chengda Biotechnology | | Released by the WHO |
| Strain, strain background (influenza virus) | H1N1 A/Brisbane/02/2018 | Chengda Biotechnology | | Released by the WHO |
| Strain, strain background (influenza virus) | H3N2 A/Kansas/14/2017 | Chengda Biotechnology | | Released by the WHO |
| Strain, strain background (influenza virus) | H3N2 A/Hong Kong/4801/2014-like | Chengda Biotechnology | | Released by the WHO |
| Biological sample (influenza virus) | H3N2 A/Hong Kong/4801/2014 | Chengda Biotechnology | | Released by the WHO |
| Antibody | MHC II (rabbit monoclonal) | BioLegend | Clone: M5/114.15.2 | 1 μg/100 μL |
| Antibody | IFN-γ (rabbit monoclonal) | BioLegend | Clone: XMG1.2; cat: 505806 | 1 μg/100 μL |
| Antibody | CD80 (Armenian hamster monoclonal) | BD Biosciences | Clone: 16-10A1; cat: 553769 | 1 μg/100 μL |
| Antibody | CD86 (rabbit monoclonal) | BD Biosciences | Clone: GL1; cat: 561962 | 1 μg/100 μL |
| Antibody | IL-17A (rabbit monoclonal) | BioLegend | Clone: TC11-18H10.1; cat: 506907 | 1 μg/100 μL |
| Antibody | IL-10 (rabbit monoclonal) | BioLegend | Clone: JES5-16E3; cat: 505005 | 1 μg/100 μL |

*Continued on next page*

*Continued*

| Reagent type (species) or resource | Designation | Source or reference | Identifiers | Additional information |
|---|---|---|---|---|
| Antibody | IL-4 (rabbit monoclonal) | BD Biosciences | Clone: H129.19; cat: 553650 | 1 µg/100 µL |
| Antibody | CD11c (Armenian hamster monoclonal) | BD Biosciences | Clone: HL3; cat: 550261 | 1 µg/100 µL |
| Antibody | IgG(H + L) (goat polyclonal) | Immunoway | Cat: RS0007 | 1:10,000 |
| Antibody | NP (rabbit polyclonal) | Sino Biological | Cat: 11675-T62 | 1:500 |
| Peptide, recombinant protein | HA of H2N2A/Canada/720/2005 | Sino Biological | Cat: 11688-V08H | |
| Peptide, recombinant protein | HA of H5N1 A/Hubei/1/2010 | Sino Biological | Cat: 40015-V08H | |
| Peptide, recombinant protein | HA of H7N9A/Shanghai/2/2013 | Sino Biological | Cat: 40239-V08B | |
| Peptide, recombinant protein | HA of H9N2 497A/HongKong/35820/2009 | Sino Biological | Cat: 40174-V08B | |
| Commercial assay or kit | Beaver Beads Protein A/G antibody Purification Kit | Beaver | Cat: 20102-1 | |
| Commercial assay or kit | Cytometric Bead Array (CBA) Mouse Immunoglobulin Isotyping Kit | GE HealthCare | Cat: 550,026 | |
| Commercial assay or kit | Mouse IL-4 ELISpot PLUS (ALP) | MabTech | Cat: 3311-4APW-2 | |
| Commercial assay or kit | Mouse IFN-gamma ELISpot PLUS (ALP) | MabTech | Cat: 3321-4APT-2 | |
| Software | GraphPad Prism | GraphPad Prism (https://graphpad.com) | RRID:SCR_015807 | Version 8.0.0 |
| Software | ImageJ | ImageJ (http://imagej.nih.gov/ij/) | RRID:SCR_003070 | Version 2.0.0 |
| Software | FlowJo | FlowJo (https://www.flowjo.com/) | RRID:SCR_008520 | Version 10 |
| Other | Sensor Chip CM5 | GE HealthCare | Cat: BR100530 | |

## Animals

Female BALB/c mice (4–6 weeks old) were purchased from Beijing Vital River Laboratory Animal Technology Co. Ltd (Beijing, China) and maintained under specific pathogen-free conditions. Feed and water were supplied ad libitum. All animal procedures were performed in accordance with Institutional Animal Care and Use Committee (IACUC) guidelines and have been approved by the IACUC of University of Chinese Academy of Sciences.

## Construction and production of recombinant NMH and NMHC proteins

The recombinant protein NMH consists of two fragments of NP (335-350aa, 380-393aa) (*Ben-Yedidia et al., 1999*), two copies of M2e (*Feng et al., 2006*), and three tandem fragments of HA2 (76-130aa from H1N1 A/California/04/2009, H3N2 A/Hong Kong/1/1968, and H7N7 A/Netherlands/219/2003) (*Wang et al., 2010*). NMHC consists of NMH at the amino terminal followed by SEC2 sequence (*Xu et al., 2008*) at the carboxyl terminal. A flexible linker peptide sequence (GSAGSAG) was designed between each component to avoid any stereo-hindrance effect. Then, the encoding DNA fragment of NMH was optimized according to the codon preference of *E. coli* and synthesized by Sangon Biotech (Shanghai, China), while the DNA fragment of NMHC was constructed via overlap polymerase chain reaction. The encoding DNA fragment were ligated into the expression vectors pET-28a (+) plasmid and transformed into *E. coli* BL21(DE3), and the positive clones were verified by DNA sequencing. Vector-containing *E. coli* strains were cultured in (Luria–Bertani) medium, and protein expression was induced with 0.5 mM isopropyl β-D-1-thiogalactopyranoside at 30°C for 8 hr. Cells were harvested and disrupted by sonication on ice bath, then supernatants and precipitates were isolated by centrifugation at 21,000 g for 15 min. SDS-PAGE electrophoresis results show that the expressed NMH and NMHC proteins were both inclusion bodies. The method of protein purification and refolding was modified from previously reported (*Lu et al., 2014*; *Song et al., 2020*). The insoluble inclusion bodies were resuspended with washing buffer (2 M urea, 50 mM Tris-HCl, 100 mM NaCl, 1 mM EDTA, pH 8.0), followed by centrifugation at 21,000 g for 15 min. This washing step was repeated twice before the washed inclusion bodies were solubilized in a denaturation buffer (8 M urea, 100 mM

NaH$_2$PO$_4$, 10 mM Tris-HCl, 20 mM imidazole, 1 mM DTT, pH 8.0) by votex-shaking at room temperature for 30 min. The supernatant was collected by centrifugation at 21,000 g for 15 min. The protein was purified by the Ni-saturated chelating sepharose affinity chromatography with the AKTA Fast Protein Liquid Chromatography System, eluting with a denaturation elution buffer (8 M urea, 100 mM NaH$_2$PO$_4$, 10 mM Tris-HCl, 250 mM imidazole, 1 mM DTT, pH 8.0). To refold the elution fractions, dialysis refolding method was used. The refolding buffer was PBS adding 0.5 M arginine, 0.2 mM oxidized glutathione, 1 mM reduced glutathione, and 4 mM EDTA supplement with 6 M, 3 M, 1.5 M, 0.75 M, and 0 M urea, respectively. The purified protein was refolded with step-wise dialysis proceeding from 6 M to 0 M urea following exchanging the refolding buffer three times with PBS.

## Recombinant protein-mediated splenocytes proliferation assay

Murine splenocytes were obtained from healthy BALB/c mice under aseptic condition as described in our previous report and maintained in RPMI 1640 medium containing 10% FBS. The freshly isolated murine splenocytes were seeded in 96-well flat-bottomed plates at $1 \times 10^6$ cells/well and stimulated for 72 hr with 3.5 µM NMHC, 3.5 µM NMH, 3.5 µM SEC2, and 3.5 µM NMH plus 3.5 µM SEC2, respectively, using PBS as negative controls. Cell proliferation was determined by MTS assay, and the PI was calculated as described previously (*Zhang et al., 2016*).

## Recombinant protein-mediated BMDCs maturation and CD4$^+$ T cell differentiation assay

Murine BMDCs were prepared as previously described (*Yao et al., 2018b*). Briefly, mice were sacrificed by intraperitoneal administration of tribromoethanol (400 mg/kg body weight), and the femur and tibia of the hind legs were dissected, then bone marrow cavities were flushed with 10 mL cold sterile PBS. After lysing red blood cells, the bone marrow cells were cultured and differentiated into BMDCs in RMPI-1640 with 10% FBS, 20 ng/mL rmGM-CSF, 10 ng/mL rmIL-4, 100 µg/mL streptomycin, and 100 U/mL penicillin. Six days later, the purity of CD11c$^+$ cells was >90% as determined by flow cytometry. Then, BMDCs were simulated for 24 hr with 3.5 µM NMH, 3.5 µM NMHC, 3.5 µM NMH plus 3.5 µM SEC2, and 3.5 µM SEC2, respectively, using PBS as negative control. The maturation of BMDCs was evaluated through detecting the cell surface markers including CD80, CD86, and MHC II with fluorescent labeled antibodies, respectively, followed by analyzing in a FACScalibur flow cytometer.

CD4$^+$ T cells were sorted from freshly isolated murine splenocytes by immunomagnetic beads. For Th cells differentiation assay, the matured BMDCs were co-cultured with CD4$^+$ T cells at a ratio of $10^4$:$10^5$ for 96 hr. Cells were fixed and permeabilized with Transcription Factor Buffer Set according to the manufacturer's instructions. Then cells were intracellular stained with fluorescent labeled antibodies against IL-4 (for Th2 subtype), IFN-γ (for Th1 subtype), IL-17A (for Th17 subtype), and IL-10 (for Treg subtype), respectively. The percentages of helper T cell subtypes were determined by flow cytometric.

## Animal immunization

Female wild-type BALB/c mice were respectively immunized with an equimolar of NMHC (200 µg), NMH (100 µg), SEC2 (100 µg), NMH (100 µg) plus SEC2 (100 µg) and PBS (control) by intraperitoneal administration at days 0 (prime), 14 (boost), and 28 (boost). Serum samples were collected on days 14, 28, 42, and 100 and kept at –80°C until use. Splenocytes were harvested at day 14 after the last immunization for ELISpot assays and flow cytometric analysis.

## Mouse immunoglobulin isotyping detected by cytometric bead assay

Isotype profiles of mouse immunoglobulins including IgA, IgE, IgG1, IgG2a, IgG2b, IgG3, and IgM were detected by Cytometric Bead Array (CBA) according to the manufacture's instruction. Shortly, for serum sample diluted 4000 times in PBS, 50 µL Mouse Ig Capture Bead Array was mixed with 50 µL standards or sera samples diluted in master buffer (1:10,000) in tube before incubation for 15 min at room temperature. Then, the beads were washed once and added with 50 µL PE/FITC detector antibody in master buffer. After incubation for 15 min at room temperature in the dark, the beads were washed once and detected by flow cytometer. The mean fluorescence intensity (MFI) of PE represented the antibody expression levels.

## Microneutralization-hemagglutinin inhibition assay (MN-HI)

Four strains of influenza virus were used for MN-HI assay to detect biological activity of serum antibody induced by recombinant protein (*Fu et al., 2016*). After treatment with receptor-destroying enzyme, serum samples were serially double diluted in 96-well plates and mixed with H1N1 A/Brisbane/02/2018 (Bris/02(H1)), H3N2 A/Kansas/14/2017 (Kan/14(H3)), H1N1 A/Michigan/45/2015 (MI/45(H1)), and H3N2 A/Hong Kong/4801/2014-like (HK/4801(H3)) at a final concentration of 100 $TCID_{50}$ (Median Tissue Culture Infectious Doses) virus per well, respectively. After incubation for 1 hr at 37°C, $1.5 \times 10^4$ MDCK cells supplemented with 2 µg/mL trypsin and 0.5% bovine serum albumin (BSA) in DMEM media were added to each well. After infection and amplification for 72 hr at 37°C, viruses were collected and incubated with Turkey Red Blood cells in 96-well plate at room temperature for 30 min. The hemagglutination status of each well was visually determined. The titers of each serum sample were defined as the reciprocal of the highest dilution where no hemagglutination was observed.

## ELISA assay

### Measurement of antigen-specific IgG by ELISA

The 96-well microtiter plates were coated with 100 µL HAs of H2N2 A/Canada/720/2005, H5N1 A/Hubei/1/2010, H7N9 A/Shanghai/2/2013, H9N2 A/Hong Kong/35820/2009, and split viruses of MI/45(H1), HK/4801(H3) at 2 mg/mL overnight. Plates were then washed three times with washing buffer (PBS with 0.05% Tween-20, pH 7.4) and blocked with 200 µL blocking buffer (PBS with 0.05% Tween-20 and 1% BSA, pH 7.4) per well for 1 hr at room temperature. After washing three times, 100 µL pre-serially diluted serum were added to each well and incubated at room temperature for 2 hr. Plates were again washed five times before adding 100 µL secondary antibody (horse radish peroxidase-labeled anti-mouse IgG, IgG1, IgG2a, 1:10,000 diluted in blocking solution) and further incubated for 1 hr at room temperature. Then, the plates were washed four times and added with 100 µL TMB substrate per well for 10 min. Enzymatic color development was stopped with 100 µL of 2 M hydrochloric acid per well, and the plates were read at an absorbance of 450 nm. Titer was defined as the highest dilution of serum antibodies at which the mean $OD_{450}$ value of the experiment group was no less than 2.1 times of the control. The experiment was repeated three times.

### Detecting the antigen-specific IL-4 and IFN-γ cytokine production by ELISA

In brief, 2 weeks after the third immunization, splenocytes from each group were collected and treated with 1000 ng/mL SEC2 and 1000 ng/mL NMH, respectively, and PBS was served as CK. Then, the culture supernatants were harvested at 48 hr to detect the production of IL-4 and IFN-γ by ELISA following the manufacturer's instructions.

## Antibody purification and pull-down assay

Serum antibodies were purified using the Beaver-Beads Protein A/G antibody Purification Kit according to the protocols provided by the manufacturer. The ability of purified Abs to form a stable complex with NMH was further confirmed in a pull-down assay (*Mallajosyula et al., 2014*). NMH and Abs were mixed together at 2:1 molar ratio and incubated for 2 hr at 4°C. The equilibrated Protein A/G beads were added to the mixture and incubated for 2 hr to bind and pull down NMH, while the unbound supernatants were separated. The antibody bound to the beads was eluted with antibody elution buffer and then neutralized with antibody neutralization buffer. The unbound and eluted fractions were subsequently analyzed by SDS-PAGE.

## Binding affinity studies using SPR

Binding affinities of the purified serum antibody with HAs of H2N2, H5N1, and split virion of H1N1, H3N2 were determined by SPR experiments performed with Biacore T200 (*Fu et al., 2016*). The purified Abs in PBSP buffer (PBS with 0.05% P20, pH 7.4) at 1.25 nM were captured onto CM5 chip through goat anti-mouse IgG (H + L) that immobilized (500–700 response units [RUs]) by standard amine coupling to the surface of the biosensors. The goat anti-mouse IgG (H + L) sensor channel served as a negative control for each binding interaction. Multiple concentrations of HAs were passed over each channel in a running buffer of PBS (pH 7.4) with 0.05% P20 surfactant. Both binding and dissociation events were measured at a flow rate of 30 µL/min. The sensor surface was regenerated

**Table 1.** Sequences for qPCR primers.

| Gene | F: forward primer (5′–3′), R: reverse primer (5′–3′) | Reference |
|---|---|---|
| | F: TGGAATCCTGTGGCATCCATGAAAC | |
| β-Actin | R: TAAAACGCAGCTCAGTAACAGTCCG | *Currier and Robinson, 2001* |
| | F: GATGTGAACCCTAGGCCAGA | |
| Perforin | R: AAAGAGGTGGCCATTTTGTG | *Currier and Robinson, 2001* |
| Granzyme B | F: ACTTTCGATCAAGGATCAGCA R: GGCCCCCAAAGTGACATTTATT | *Han et al., 2010* |
| | R: GGCCCCCAAAGTGACATTTATT | |
| IFN-γ | F: AGACAATCAGGCCATCAGCA | |
| | R: TGGACCTGTGGGTTGTTGAC | *Yao et al., 2018a* |
| IL-4 | F: GAGACTCTTTCGGGCTTTTCG | |
| | R: CAGGAAGTCTTTCAGTGATGTGG | *Chen et al., 2012* |
| T-bet | F: ATTGCCCGCGGGGTTG | |
| | R: GACAGGAATGGGAACATTCGC | *Chen et al., 2012* |
| GATA-3 | F: GGTCAAGGCAACCACGTC | |
| | R: CATCCAGCCAGGGCAGAG | *Chen et al., 2012* |
| H1(HA) | F: CAGATTYTGGCGATCTAYTC | |
| | R: GACCCATTAGARCACATCCAG | |

after every binding event by repeated washing with glycine pH 2.0. Each binding curve was analyzed after correcting for nonspecific binding by subtraction of signal obtained from the negative control flow channel. The KD values were calculated using a steady affinity state model by the BIAcore T200 evaluation software (version 3.1) (*Zhang et al., 2013*).

## ELISpot assays

Splenocytes were freshly isolated from immunized mice of each group. Influenza-specific IgG ASCs were enumerated with ELISpot. MultiScreen HTS 96-well plates were coated with purified NMH at a concentration of 5 µg/mL in PBS to detect antigens specific ASCs, or coated with split viruses of H1N1 and H3N2 at a concentration of 5 µg/mL in PBS to detect influenza virus-specific ASCs. Wells coated with PBS served as negative controls. Plates were incubated overnight at 4°C and blocked with complete medium for 2 hr at 37°C. Then splenocytes were added into triplicate wells (500,000 cells/well) and incubated at 37°C for 20 hr. Plates were vigorously washed to remove cells, and then incubated with HRP-conjugated anti-mouse IgG overnight at 4°C. The spots representing the IgG ASCs in each well were counted by inspection. Nonspecific spots detected in the negative control (PBS) wells were subtracted from the counts of each total ASCs. Moreover, ELISpot kits were used to measure the IFN-γ or IL-4-producing splenocytes on site according to the manufacturer's protocol.

## Real-time quantitative polymerase chain reaction (qPCR)

To evaluate the expressions of inflammatory cytokines and transcription factors in splenocytes and to detect the virus titers in lungs of infected mice, total RNA were extracted from splenocytes or lung tissues using the RNA-extracting reagent RNAiso Plus, and 1 µg of total RNA were reverse transcribed to cDNA using a PrimeScript RT Master Kit according to the manufacturer's instructions. Resulting cDNA was used for qPCR analysis with a SYBR Green PCR kit (Roche). β-Actin was used as reference gene. All primers are listed in *Table 1*. Relative transcription levels were determined using the $2^{-\Delta\Delta Ct}$ analysis method (*Fu et al., 2020*).

## Cytotoxic lysis assay

In vivo cytotoxic lysis assay was performed as previously reported (*Xie et al., 2014*). Splenocytes from naïve BALB/c mice were divided into two parts. One part was pulsed with $10^{-6}$ M NMH peptides and labeled with 10 µM of CFSE (termed as CFSE$^{high}$ target cells). The other part was pulsed with $10^{-6}$ M BSA and labeled with 1 µM of CFSE (termed as CFSE$^{low}$ cells) as a non-target control. Cells from the two parts were mixed in a 1:1 ratio and injected into immunized recipient mice at $2 \times 10^7$ total cells per mouse via the tail vein on day 14 after the third immunization. Twelve hours later, splenocytes were isolated from the recipients and differential CFSE fluorescent intensities were measured with a flow cytometry. Specific lysis was calculated using the following formula: Percentage of specific lysis = (1 – [ratio unprimed/ratio primed] × 100), where ratio = percentage CFSE$^{low}$/percentage CFSE$^{high}$.

## Challenge experiments

The challenge experiments were performed as previously reported (*Meng et al., 2013*). BALB/c mice were immunized with 200 µg NMHC, 100 µg NMH alone, 100 µg NMH plus 100 µg SEC2, or PBS. Two weeks after the third immunization, mice were challenged with virus. Before virus infection, 12 mice of each group were anesthetized and inoculated intranasally with $10^3$ TCID$_{50}$ of H1N1 MI/45(H1) and H1N1 Bris/02(H1) virus strains in a volume of 50 µL. To determine the viral load and the pathological damage in the infected lungs, three mice from each group were sacrificed on days 3 and 6 post-infection. As previously reported (*Meng et al., 2013*; *Marcos et al., 2017*), the right lungs were used for qPCR to assess the virus titer, while the left lungs were fixed for the histopathological analysis. Survival and weight change of the remaining mice in each group were monitored daily for 14 days after the infection.

## Histological analysis

The mice were sacrificed on days 3 and 6 after infection with virus. The left lungs were collected and dissected for histological observation (n = 3 mice per group). After fixation in neutral-buffered fixative, the tissues were embedded in paraffin and stained with hematoxylin and eosin. The lungs were sliced into 6-µm-thick frozen sections, and the lung-infiltrating leukocytes profiles in the tissues were reflected by IHC analysis of CD45. The distribution of influenza viruses, lung-infiltrating T lymphocytes, and lung-infiltrating macrophages was respectively reflected by immunofluorescence analysis of NP, CD8, and F4/80 with Laser Scanning Confocal Microscope (LSCM).

## Statistical analysis

The data were analyzed by Student's t-test, one-way analysis of variance (ANOVA), and followed by a suitable post-hoc test using the SPSS 26.0 and GraphPad Prism Software (version 6.0c). Differences with p-values<0.05 were considered to be statistically significant.

## Acknowledgements

We thank J Ludovic Croxford, PhD, from Liwen Bianji (Edanz) (https://www.liwenbianji.cn) for editing the language of a draft of this manuscript.

## Additional information

### Competing interests

Libao Zhou, Hui Liao, Songyuan Yao: Is an employee of Chengda Biotechnology Co. Ltd. The author declares that no other competing interests exist.. The other authors declare that no competing interests exist.

## Funding

| Funder | Grant reference number | Author |
| --- | --- | --- |
| Strategic Priority Research Program of the Chinese Academy of Sciences Grant | XDA12020225 | Mingkai Xu |
| Liaoning Revitalization Talents Program | XLYC1807226 | Mingkai Xu |
| Science and Technology Plan Projects of Shenyang City Grants | Y17-4-003 | Mingkai Xu |
| Shengyang High level Innovative Talents Program | RC190060 | Mingkai Xu |
| Science and Technology Agency Livelihood Program of Liaoning Province of China | 2021JH2/10300075 | Mingkai Xu |

The funders had no role in study design, data collection and interpretation, or the decision to submit the work for publication.

## Author contributions

Yansheng Li, Data curation, Formal analysis, Investigation, Methodology, Software, Writing – original draft; Mingkai Xu, Conceptualization, Funding acquisition, Project administration, Supervision, Writing – review and editing; Yongqiang Li, Gulinare Halimu, Methodology, Software; Wu Gu, Data curation, Investigation; Yuqi Li, Formal analysis, Software; Zhichun Zhang, Songyuan Yao, Methodology; Libao Zhou, Hui Liao, Methodology, Resources; Huiwen Zhang, Investigation, Methodology; Chenggang Zhang, Formal analysis, Investigation

## Author ORCIDs

Mingkai Xu http://orcid.org/0000-0001-9889-6287

## Ethics

This study was performed in strict accordance with the recommendations in the Guide for the Care and Use of Laboratory Animals of the National Institutes of Health . All animal procedures were performed according to approved Institutional Animal Care and Use Committee (IACUC) guidelines of University of Chinese Academy of Sciences (permit-number IAE.No20191010A0120). All surgery was performed under tribromoethanol anesthesia, and every effort was made to minimize suffering.

## Decision letter and Author response

Decision letter https://doi.org/10.7554/eLife.71725.sa1
Author response https://doi.org/10.7554/eLife.71725.sa2

---

# Additional files

## Supplementary files

• Supplementary file 1. Breadth of the antibody response elicited by the recombinant proteins. (a) Neutralizing antibody titers detected by standard HI. Sera were collected from immunized mice on day 42 after immunization, and the neutralization assays were performed against MI/45(H1), HK/4801(H3), Influenza Virus Infectious NYMC BX-35 (Victoria), B/Phuket/3073/2013-like virus (Yamagata) influenza viruses. The hemagglutination status of each well was visually determined. The titers of each serum sample were defined as the reciprocal of the highest dilution where no hemagglutination was observed. (b) ELISA endpoint titers of HAs or split virion. On day 42 after immunization, breadth of the antibody response elicited by the recombinant protein NMHC was determined by ELISA of the pooled antisera against purified rHA proteins or split virion. Titer was defined as the highest dilution of serum antibodies at which the mean $OD_{450}$ value of the experiment group was no less than 2.1 times of the control.

• Transparent reporting form

## Data availability

All data generated or analysed during this study are included in the manuscript and supporting files.

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
