## [Editor Report]

The authors provided a new concept for making a universal influenza vaccine by fusing T cell epitopes, B cell epitopes, and superantigen. They conclude that their construct is effective against broad influenza strains, and now the convincing evidence for their conclusion has been provided.

---

## [Decision Letter]

**Decision letter after peer review:**

Thank you for submitting your article "A recombinant protein containing influenza viral conserved epitopes and superantigen induces broad-spectrum protection" for consideration by *eLife*. Your article has been reviewed by 3 peer reviewers, one of whom is a member of our Board of Reviewing Editors, and the evaluation has been overseen by Tadatsugu Taniguchi as the Senior Editor. The following individual involved in review of your submission has agreed to reveal their identity: Tony Moody (Reviewer #3).

Essential revisions:

1) The authors have immunized mice three times every two weeks. The usual current influenza vaccine is immunized twice against mice to test its efficacy. The authors should discuss the comparison of the effectiveness of the current vaccine and theirs.

2) They conducted a cytometric bead array to calculate immunoglobulin subclasses in sera presented in Figure 3. In the control group on Day 100, the MFI of IgG2b is reduced compared to other time points. They should calculate the absolute amount of IgG2b, not MFI, by using standard.

3) The data on neutralization assays in Figure 4 is not sufficiently convincing. The authors should perform neutralization assays for H1N1 and H3N2 and other strains to prove that their vaccine candidate is effective against a wide range of influenza viruses. Also, they should conduct a statistical test to show significant differences.

4) The ELISA and Biacore assay indicate that the binding capacity of the antibodies induced by vaccination is higher against H2, H5, and H9 than H1, H3, and H7. They should discuss this.

5) Examining T cell responses is very important. The T-cell responses induced by NMHC are not different from those induced by SEC2 alone in Figure 8. Hence, the authors should verify whether the induced T cell response is indeed influenza antigen-specific or not. Specifically, cytokine production should be detected after stimulating splenocytes of post-immunized mice with influenza antigens without SEC2 and SEC2 alone.

*Reviewer #1:*

To make cross-protective vaccines for influenza, authors have made recombinant proteins composed of influenza conserved epitopes and super-antigen and examined. Particularly, they seemed to wish to show up the importance of fusion of super-antigen and viral antigen, although authors already demonstrated this concept in a previous paper by using another system.

Ab and T cell responses were analyzed which is good. But, every data are superficial; for instance, authors have not examined which epitopes Abs recognize, and which epitopes T cell recognize. Hence, this study does not have a significant impact.

In this study, Li and colleagues generated a novel recombinant protein NMHC as a potential candidate of universal broad-spectrum vaccine for various influenza virus prevention and therapy. The NMHC protein consisted of two domains. One is NMH domain consisting of influenza viral highly conserved long α-helix regions of HA from group 1 and group 2, NP (two epitopes of CTL and helper T lymphocytes) and M2e fragments as universal immunogen. The other is SEC2 domain as adjuvant-like molecular chaperone for enhancing the antigenicity of NMH. This vaccine strategy sounds interesting and promising and it certainly seemed that SEC2 domain increased ability of mice to protect against influenza viruses by enhancing the immune response for NMH domain. However, as shown below, in some respects, there seemed to be a lack of data and interpretation about the significance of incorporating all the elements as one domain.

I agree that the NP and M2 sequences are highly conserved across influenza virus subtypes, but it is necessary to validate the conservativeness more carefully about the peptide sequences used in NMH domain with the information of predicted binding ability to MHC (class I or II) by a tool like NetMHC. In particular, since the T epitope is greatly affected by even one amino acid difference, so please compare the typical influenza virus strains from H 1 (2009 pandemic type), H1 (Russian type or PR8), H2, H5, H3, H7, and H9 subtypes. This information seems to be important in considering the breadth of vaccine effect. If authors think HA2 element also includes conserved T cell epitopes covering various HA subtypes, please analyze together.

Probably it's being evaluated as an initial basic data, but it seems important to mention the author's interpretation about the importance of NP and Me2 elements in NMH protein. Especially, if authors think these elements also have critical role, it is necessary to show the data how NP or Me2 elements in NMHC protein work effectively for immune response by comparing with NP or Me2 elements excluded protein. In particular, these data seem to be important for considering the specific antigen for T cell-B cell interaction in Figure 7 and the antigen specificity of CD4 T cells activated by NMHC immunogen in Figure 8.

The authors mentioned that HA2 element in NMH domain is important for the induction of broadly cross-reactive neutralizing antibodies. However, antibodies or memory/GC B cells induced by NMHC immunization have not been evaluated at the single clone or single cell level. In particular, if authors think that connecting of each long α-helix region from H1, H3 and H7 HA in tandem as one domain has some advantages, for example it will increase the number of clones that can react to both grou1 and group2 strains, the experimental data is necessary. Anyway because the lacking of the data and interpretation for this point, it is difficult to evaluate the broadness of cross-reactive antibody induced by HA2 element in NMH domain. At least I would like authors to explain more carefully about this point (broadly cross reactivity at the level of single clone of antibody or single cell).

Regarding Figure 4 and 10-12. The authors mentioned in discussion part that it is very hard to use other strains except some restricted ones in your institute. However, H1N1 strains used in these figures are basically very closed to 2009 pandemic California strain used in NMH domain. If you want to show the antibodies induced recombinant proteins have broadly cross reactivity and protection, it is important to test the strains far from 2009 pandemic strains like representative laboratory strains PR8(H1N1) or RG14 (H5N1 *BSL2 strain). If it is impossible to use even these kinds of laboratory strains, authors should design the H1 α-helix regions in NMH domain from distant strains like H2, H5 or PR8(H1). On the other hand, in the case of H3N2 A/Hong Kong/4801/2014 strain, it is away from A/Hong Kong/1/1968(This HA is used in X31 laboratory strain). So it seems to be enough.

Regarding Figure 8. In this manuscript, authors did not touch the germinal center story like GC B cells or Tfh cells at all but these kinds of cells are very important in considering humoral immunity like affinity maturation of broadly cross reactive antibodies. So some information about Tfh marker like Bcl6, IL21 or interpretation about the germinal center reaction seem important at least even if it is difficult to perform detailed flow cytometry analysis.

*Reviewer #2:*

This paper aims to provide a novel universal influenza vaccine that could induce broad protection against various influenza viruses. For this purpose, the authors designed a new vaccine candidate who is composed of the conserved fragments of nucleoprotein (two epitopes of CTL and helper T lymphocytes), matrix protein 2, highly conserved long α-helix regions of hemagglutinin from group 1 (H1), and group 2 (H3 and H7), and superantigen Staphylococcal Enterotoxin C2, which is named NMHC. They showed that NMHC induces not only antibody production but also T-cell responses. The induced antibodies could bind broad rHA proteins and neutralize H1N1 1A/Michigan/45/2015 (group 1) and H3N2 A/Hong Kong/ 4801/2014 (group 2) viruses.

Their proposal is potentially interesting. However, the quality of the results is not sufficient enough to support their conclusion. They concluded that NMHC is a potential candidate for the universal broad-spectrum vaccine for various influenza virus prevention, whereas few data about protection against strains other than H1N1 and H3N2. In addition, there is no comparison to the effectiveness of current vaccines. Neutralization assays not only for H1N1 and H3N2 but also for other strains would strengthen their proposal. If it is difficult to use alive viruses for this purpose, it is recommended to use a simple experimental system such as pseudotyped viruses. For CD4^+^ T cell analysis, various cytokines were induced even in superantigen alone. The authors should show the specificity for T-cell responses against influenza antigens.

1) The authors have immunized mice three times every two weeks. The usual current influenza vaccine is immunized twice against mice to test its efficacy. The authors should discuss the comparison of the effectiveness of the current vaccine and theirs.

2) They conducted a cytometric bead array to calculate immunoglobulin subclasses in sera presented in Figure 3. In the control group on Day 100, the MFI of IgG2b is reduced compared to other time points. They should calculate the absolute amount of IgG2b, not MFI, by using standard.

3) The data on neutralization assays in Figure 4 is not sufficiently convincing. The authors should perform neutralization assays for H1N1 and H3N2 and other strains to prove that their vaccine candidate is effective against a wide range of influenza viruses. Also, they should conduct a statistical test to show significant differences.

4) The ELISA and Biacore assay indicate that the binding capacity of the antibodies induced by vaccination is higher against H2, H5, and H9 than H1, H3, and H7. They should discuss this.

5) The T-cell responses induced by NMHC are not different from those induced by SEC2 alone in Figure 8. The authors should verify whether the induced T cell response is influenza antigen-specific. Specifically, cytokine production should be detected after stimulating splenocytes of post-immunized mice with influenza antigens without SEC2.

6) Two IgG2b are shown in the Plot representing the standard in Figure 3. I guess that one of the two is IgG2a. Please correct the figure.

7) There is no match between each KD shown in Figure 6 and the KD in the text.

8) The sentence "IL-4 (for Th1 subtype), IFN-γ (for Th2 subtype)" should be corrected to "IFN-γ (for Th1 subtype), IL-4 (for Th2 subtype)" (Page 34, Line 558).

*Reviewer #3:*

This manuscript describes a vaccine construct comprised of multiple segments derived from influenza proteins, and in some cases conjugated to staphylococcal enterotoxin C2. There is an impressive amount of data and the workup of the immune response is quite detailed, including humoral, cellular, and protection data in mice. Overall, the impression is that this construct has the potential to move forward as a vaccine candidate. In addition, the provided data support their conclusions.

The data are clearly presented in most sections, and I am impressed that in places where multiple replicates were performed, the authors listed the number of replicates and the statistical measures.

The grammar of the manuscript needs work, and the text would benefit from copy-editing at a minimum and might even benefit from editing for length in some sections.

In some places, I would probably have presented some of the data differently. For example, I typically display bead-based antibody binding assays on a logarithmic scale which is more in line with the type of data shown. That being said, there is nothing deceptive in how the data are presented, so a change may not be necessary.

I do not recommend additional experimentation. I do think that moving this candidate forward through regulatory bodies may face significant hurdles, but that is not the problem being addressed in this paper.

Finally, I think Figure 13 is really more suited to a review article and not appropriate for this manuscript.

---

## [Author Response]

Essential revisions:1) The authors have immunized mice three times every two weeks. The usual current influenza vaccine is immunized twice against mice to test its efficacy. The authors should discuss the comparison of the effectiveness of the current vaccine and theirs.

Thanks for your careful reading and constructive advices. In fact, in our preliminary experiment, we have optimized the dosage and immune frequency of NMHC by detecting the H1N1 and H3N2 specific IgG by ELISA. Based on the result in Author response table 1, we chose a dose of 200 μg each time and each mice for three immunizations in order to achieve a better immune effect. The main objective of this study is to evaluate the feasibility of this recombinant vaccine strategy, and three immunizations for recombinant influenza vaccine have been frequently reported by researchers such as Joan E. M (doi:10.1038/s41541-018-0063-7), Raffael Nachbagauer (doi:10.1038/npjvaccines.2016.15) and John Steel (doi: 10.1128/mBio.00018-10). As you mentioned, the usual current influenza vaccine is immunized twice against mice. Most of the usual current influenza vaccines are inactivated quadrivalent influenza vaccines (QIV) which are more immunogenic than the recombinant vaccine. So two immunizations might be enough for QIV but not for recombinant vaccine. As shown in the supplementary materials (Table S1), the neutralizing antibodies induced by QIV could be effectively detected by HI test but not induced by the recombinant vaccine.

**Author response table 1. sa2table1:** The antibody titers in preliminary experiment.

Treatment groups(μg)	Serum antibody titer(H1N1)	Serum antibody titer(H3N2)				
200 NMHC 3rd	>2048	>2048	>2048	1024	1024	1024
100 NMHC 3rd	1024	>2048	1024	256	256	256
200 NMHC 2nd	1024	1024	1024	256	512	256
100 NMHC 2nd	512	1024	512	256	128	128

2) They conducted a cytometric bead array to calculate immunoglobulin subclasses in sera presented in Figure 3. In the control group on Day 100, the MFI of IgG2b is reduced compared to other time points. They should calculate the absolute amount of IgG2b, not MFI, by using standard.

Thanks for your careful reading and constructive advices. According to the instruction manual of Mouse Immunoglobulin Isotyping Kit (Cat. No.550026), we used the MFI, the original data, to represent the amount of immunoglobulins. The instruction manual of this kit did not provide how to calculate the absolute amount of each immunoglobulin. Furthermore, MFI was widely used by researchers such as Edward Morgan (doi:10.1016/j.clim.2003.11.017), Yunxiao Zhao (doi:10.1016/j.cca.2014.09.018) and Yang He (doi:10.1016/j.cca.2012.10.035) to represent the amount of immunoglobulins. As your advice, we tried to calculate the absolute amount of IgG2b by using standards, which contain 0.25 μg/ml each of IgG1 κ/λ, IgG2a κ/λ, IgG2b κ/λ, IgG3 κ/λ, IgA κ/λ, IgM κ/λ and IgE κ, and the absolute amount of IgG2b was calculated using the following equation:

MFI IgG2bκ unknown sample/MFI IgG2bκ standard * the dilution multiple *0.25μg/ml

The results showed in the following Author response table 2, the trend is similar to MFI.

**Author response table 2. sa2table2:** The absolute amount of IgG2bκ.

	NMHC	NMH	SEC2	NMH^+^SEC2	PBS
day 100IgG2bκ (mg/ml)	2.53±0.087	2.483±0.144	2.24±0.104	2.277±0.116	1.393±0.006

3) The data on neutralization assays in Figure 4 is not sufficiently convincing. The authors should perform neutralization assays for H1N1 and H3N2 and other strains to prove that their vaccine candidate is effective against a wide range of influenza viruses. Also, they should conduct a statistical test to show significant differences.

Thanks for your comments and constructive advices. According to your suggestion, we have supplemented neutralization assays for H1N1 A/Brisbane/02/2018 and H3N2 A/Kansas/14/2017 using the same batch of frozen serum samples which were used in neutralization assays for H1N1 A/Michigan/45/2015 and H3N2 A/Hong Kong/ 4801/2014 in our original manuscript. These four strains are all the living influenza virus strains we can obtain in current situation. We are so sorry that we have not been allowed to obtain any other living influenza virus strains to further verify the broad spectrum of NMHC. We agree that neutralization assays for these four strains were not enough to say that NMHC was effective against a wide range of influenza viruses. Therefore, we used the ELISA and Biacore assays, selected the HA fragments of representative influenza virus subtypes, to evaluate the broad binding properties of antibodies induced by NMHC. According to your advice, we reanalyzed the results of neutralization assays and conducted a statistical test to show significant differences by geometric mean titer (GMT). In our revised manuscript, the original Figure 4 has been replaced by the new Figure 4.

4) The ELISA and Biacore assay indicate that the binding capacity of the antibodies induced by vaccination is higher against H2, H5, and H9 than H1, H3, and H7. They should discuss this.

Thanks for your careful reading and comment. As described in Materials and methods part in our manuscript, in the ELISA and Biacore assays, H2, H5, H7, and H9 were commercial recombinant HA proteins, while H1 and H3 were split viruses. It might be that the complex composition and conformation of the split viruses led to the relatively lower binding capacities to H1 and H3 in the ELISA and Biacore results. As to H7, we can see from the Table S2, the ELISA endpoint titers calculated from Figure 5, showed that the titer of H7 induced by NMHC was the same with titers of H2, H5, and H9.

Theoretically, the long α-helix regions of hemagglutinin (amino acids 76–130 from the HA2) were conserved among H1, H2, H3, H5, H7, and H9 (Author response image 1), so the binding capacities of antibodies induced by NMHC to these HAs should be similar. Furthermore, we learned a lot from your advices and we really appreciate your professional comments.

**Author response image 1. sa2fig1:** The long α-helix amino acids 76–130 from the HA2 of different hemagglutinin subtypes.

5) Examining T cell responses is very important. The T-cell responses induced by NMHC are not different from those induced by SEC2 alone in Figure 8. Hence, the authors should verify whether the induced T cell response is indeed influenza antigen-specific or not. Specifically, cytokine production should be detected after stimulating splenocytes of post-immunized mice with influenza antigens without SEC2 and SEC2 alone.

Thank you very much for your suggestion. As you mentioned, the T-cell responses in figure 8 was not antigen-specific because that the T cells had not been treated with any antigens in vitro. According to your advice, we have repeated the animal immunization experiment to detect the IL-4 and IFN-γ cytokine production after stimulating splenocytes of post-immunized mice with NMH without SEC2 and SEC2 alone. In brief, female wild-type BALB/c mice were respectively immunized with 200 μg NMHC, 100 μg NMH, 100 μg SEC2, 100 μg NMH plus 100 μg SEC2, and PBS (control) by i.p. for three times with interval of 14 days. Two weeks after the third immunization, splenocytes from each group were collected and treated with 1000 ng/mL SEC2 alone or 1000 ng/mL NMH alone, and PBS was served as CK. Then, the culture supernatants were harvested at 48 h to detect the production of IL-4 and IFNγ by ELISA following the manufacturer’s instructions. In Figure 8C the results showed that, in NMHC, NMH^+^SEC2 and NMH immunized groups, the productions of IL-4 and IFN-γ treated with NMH were significantly higher than CK (*p* < 0.05), while SEC2 and PBS immunized groups didn't show significant changes. This result verified that the T cell response induced by NMH treatment is influenza antigen-specific. However, the splenocytes from all of the immunized groups produced high levels of IL-4 and IFN-γ after SEC2 treatment in vitro, and there were no significant differences among the five immunized groups. As a superantigen, SEC2 could induce nonspecific T cell responses, leading to massive production of cytokine. According to your comment, we have added this supplementary experiment and results to our revised manuscript, and added to Figure 8C in our revised manuscript.

Reviewer #1:To make cross-protective vaccines for influenza, authors have made recombinant proteins composed of influenza conserved epitopes and super-antigen and examined. Particularly, they seemed to wish to show up the importance of fusion of super-antigen and viral antigen, although authors already demonstrated this concept in a previous paper by using another system.Ab and T cell responses were analyzed which is good. But, every data are superficial; for instance, authors have not examined which epitopes Abs recognize, and which epitopes T cell recognize. Hence, this study does not have a significant impact.In this study, Li and colleagues generated a novel recombinant protein NMHC as a potential candidate of universal broad-spectrum vaccine for various influenza virus prevention and therapy. The NMHC protein consisted of two domains. One is NMH domain consisting of influenza viral highly conserved long α-helix regions of HA from group 1 and group 2, NP (two epitopes of CTL and helper T lymphocytes) and M2e fragments as universal immunogen. The other is SEC2 domain as adjuvant-like molecular chaperone for enhancing the antigenicity of NMH. This vaccine strategy sounds interesting and promising and it certainly seemed that SEC2 domain increased ability of mice to protect against influenza viruses by enhancing the immune response for NMH domain. However, as shown below, in some respects, there seemed to be a lack of data and interpretation about the significance of incorporating all the elements as one domain.I agree that the NP and M2 sequences are highly conserved across influenza virus subtypes, but it is necessary to validate the conservativeness more carefully about the peptide sequences used in NMH domain with the information of predicted binding ability to MHC (class I or II) by a tool like NetMHC. In particular, since the T epitope is greatly affected by even one amino acid difference, so please compare the typical influenza virus strains from H 1 (2009 pandemic type), H1 (Russian type or PR8), H2, H5, H3, H7, and H9 subtypes. This information seems to be important in considering the breadth of vaccine effect. If authors think HA2 element also includes conserved T cell epitopes covering various HA subtypes, please analyze together.

It have been extensively studied that both NP and M2e fragments are conserved antigens inducing cellular or humoral immune responses (Tamar Ben-Yedidia et al., DOI: 10.1016/j.vaccine.2005.08.061, Neirynck et al. DOI: 10.1038/13484, Marina et al. DOI: 10.1016/j.vaccine.2005.08.061). In our study, NMH also contained the NP and M2e elements, and the conservativeness about the peptide sequences used in NMH domain has been validated using the online computational MHC binding prediction tool called TepiTool (http://tools.iedb.org/tepitool/.). All peptide lengths (8-14) were predicted for their binding affinity to 6 alleles of MHC I and 3 alleles of MHC II. A panel of 6 MHC I (H-2-Db, H-2-Dd, H-2-Kb, H-2-Kd, H-2-Kk, H-2-Ld) and a panel of 3 MHC II (H-2-IAb, H-2-IAd, H-2-IEd) were selected. As default recommended settings, peptide selection criterion were Cutoff of IC50 ≤ 500 nM (MHC I) and Cutoff of predicted percentile rank ≤ 10% (MHC II). The results (Author response image 2) showed that the NP and M2 sequences used in NMH domain could bind to MHC class I or II. Subsequently, we also analyzed the T cell epitopes of HA2 elements using the same methods and parameters, and the results showed that the HA2 sequences used in NMH domain could bind to MHC class I or II (Author response image 3).

**Author response image 2. sa2fig2:** The information of predicted binding ability to MHC (class I or II) for the NP and M2 peptide sequences. **(A**), the sequences that were used in the binding prediction. The amino acid residues 1 to 16 were NP335–350; 17 to 30 were NP380–393; 31 to 37 were linker; 38 to 83 were 2*M2e. (**B**) and (**C**), for MHC I and MHC II, the sequence number of the source protein, start and end coordinates of the peptide within the source protein, peptide sequence, allele, IC50, and Consensus percentile rank.

**Author response image 3. sa2fig3:** The information of predicted binding ability to MHC (class I or II) for the HA2 peptide sequences. (**A**), the sequences that were used in the binding prediction. The amino acid residues 1 to 55, 63 to 117, 125 to 179 respectively were HA2 of H1, H3 and H7; 56 to 62 and 118 to 124 were linker. (**B**) and (**C**), for MHC I and MHC II, the sequence number of the source protein, start and end coordinates of the peptide within the source protein, peptide sequence, allele, IC50, and Consensus percentile rank.

As we known, MHC class I molecules have a binding groove that limits the size of its ligands to roughly 8 to 11 residues in length, while Class II molecules allowed them to bind longer peptides, typically 12 to 20 residues in length. The amino acid sequence of a peptide to MHC molecule is an important factor that determines potential immunogenicity related to the breadth of vaccine effect. Therefore, we compared the NP and M2e of NMH with typical influenza virus strains from A/Arkhangelsk/CRIE-109/2016(H1N1), A/Alabama/01/2010(H1N1), A/Panama/1/66(H2N2), A/Akita/4/1993(H3N2), A/Bangladesh/3233/2011(H5N1), A/Anhui/1-DEWH730/2013(H7N9) and A/Beijing/1/2016(H9N2) subtypes. The results (Author response image 4) showed that the conservation of NP335-350, NP380-393 and M2e were 87.5%, 100% and 94.53%, respectively, indicating the high conservation of T epitopes.

**Author response image 4. sa2fig4:** The conservation of NP and M2e sequence of different influenza virus subtypes. The sequence of H1N1, H2N2, H3N2, H5N1, H7N9, H9N2 subtypes and NMH (red frame) are listed in single amino acid code from top to bottom. Observed replacement mutations are listed beneath by single amino acid code.

Probably it's being evaluated as an initial basic data, but it seems important to mention the author's interpretation about the importance of NP and Me2 elements in NMH protein. Especially, if authors think these elements also have critical role, it is necessary to show the data how NP or Me2 elements in NMHC protein work effectively for immune response by comparing with NP or Me2 elements excluded protein. In particular, these data seem to be important for considering the specific antigen for T cell-B cell interaction in Figure 7 and the antigen specificity of CD4 T cells activated by NMHC immunogen in Figure 8.

The roles of conserved epitopes in influenza A virus, especially NP and M2e, in design of broad-spectrum influenza vaccine have been widely studied (Wang Wenling et al., doi: 10.1371/journal.pone.0052488; Raffael Nachbagauer et al., doi:10.1038/npjvaccines; Taia T. Wanga et al., DOI: 10.1073/pnas). In our manuscript, NMH and NMHC are designed based on these previous studies. We think that both NP and M2e are surely important. We are very sorry that we did not compare the results of NP or M2e excluded protein. NMHC consists of NP, M2e, three different HA2 fragments and SEC2 connected by flexible linker (GSAGSAG).

Our main aim is to explore whether the vaccine strategy is feasible. The structural modeling of NMHC in silico revealed that the elements of NP, M2e, HA2(H1, H3, H7) and SEC2 were independent from each other (Figure 1D in the manuscript), and these immunogens could play their respective functions. We are more concerned about the role of SEC2 in enhancing the protective effect induced by NMH. As showed in Figure 7 and Figure 8 in our revised manuscript, the introduction of SEC2 enhanced the T cell-B cell interaction and the antigen specificity of CD4 T cells activated by NMH. We agree that it is important to show the data how NP, M2e or HA2 elements in NMHC protein work effectively for immune responses. However, it is hard to test every element or any combination of these elements deeply in the limited time and space.

The authors mentioned that HA2 element in NMH domain is important for the induction of broadly cross-reactive neutralizing antibodies. However, antibodies or memory/GC B cells induced by NMHC immunization have not been evaluated at the single clone or single cell level. In particular, if authors think that connecting of each long α-helix region from H1, H3 and H7 HA in tandem as one domain has some advantages, for example it will increase the number of clones that can react to both grou1 and group2 strains, the experimental data is necessary. Anyway because the lacking of the data and interpretation for this point, it is difficult to evaluate the broadness of cross-reactive antibody induced by HA2 element in NMH domain. At least I would like authors to explain more carefully about this point (broadly cross reactivity at the level of single clone of antibody or single cell).

It has been widely proved that the HA2 element could induce broadly cross-reactive neutralizing antibodies. We connected the highly conserved long α-helix regions from H1, H3 and H7 HA2 in tandem in order to increase the immunogenicity. However, we don't think this combination is perfect, and the immune responses may change as the HA2 elements sequence and quantity changes. The recombinant protein NMHC is rather an example for this vaccine design strategy than a certain vaccine. The structure of NMHC must be further optimized before it become a candidate vaccine. Here, we initial studied the specific antibody secreting cells responses of HA2 (H1-H3-H7) on NMH, H1N1 and H3N2 with ELISPOT assay (Manuscript Figure 7). Furthermore, we also evaluated the broadness of cross-reactive antibody induced by HA2 element in NMH domain with the ELISA (Manuscript Figure.5) and Biacore (Manuscript Figure.6) assay. In conclusion, these results showed that the combination strategy is feasible, because HA2 elements in NMH domain could induce the broadness of cross-reactive antibody. We are so sorry that we did not supplement the single clone or single cell experiments in the limited time and space available.

Regarding Figure 4 and 10-12. The authors mentioned in discussion part that it is very hard to use other strains except some restricted ones in your institute. However, H1N1 strains used in these figures are basically very closed to 2009 pandemic California strain used in NMH domain. If you want to show the antibodies induced recombinant proteins have broadly cross reactivity and protection, it is important to test the strains far from 2009 pandemic strains like representative laboratory strains PR8(H1N1) or RG14 (H5N1 *BSL2 strain). If it is impossible to use even these kinds of laboratory strains, authors should design the H1 α-helix regions in NMH domain from distant strains like H2, H5 or PR8(H1). On the other hand, in the case of H3N2 A/Hong Kong/4801/2014 strain, it is away from A/Hong Kong/1/1968(This HA is used in X31 laboratory strain). So it seems to be enough.

Thanks for your comments and constructive advices. According to your suggestion, we have supplemented neutralization assays for H1N1 A/Brisbane/02/2018 and H3N2 A/Kansas/14/2017 using the same batch of frozen serum samples which were used in neutralization assays for H1N1 A/Michigan/45/2015 and H3N2 A/Hong Kong/ 4801/2014 in our original manuscript (Figure 4). These four strains are all the living influenza virus strains we can obtain in current situation. We are so sorry that we have not been allowed to obtain any other living influenza virus strains to further verify the broad spectrum of NMHC. We agree that neutralization assays for these four strains were not enough to say that NMHC was effective against a wide range of influenza viruses. Therefore, we used the ELISA and Biacore assay, selected the HA fragments of representative influenza virus subtypes, to evaluate the broad binding properties of antibodies induced by NMHC. In our revised manuscript, the original Figure 4 has been replaced by the new Figure 4.

Regarding Figure 8. In this manuscript, authors did not touch the germinal center story like GC B cells or Tfh cells at all but these kinds of cells are very important in considering humoral immunity like affinity maturation of broadly cross reactive antibodies. So some information about Tfh marker like Bcl6, IL21 or interpretation about the germinal center reaction seem important at least even if it is difficult to perform detailed flow cytometry analysis.

Thanks for your comments and constructive advices. In our revised manuscript, Figure 7 showed the specific antibody secreting B cells response to H1N1, H3N2 and NMH, and Figure 8 showed the specific Th cells response to NMH. As you indicated, GC B cells and Tfh cells are very important in evaluating the broadly cross-protective antibody response induced by vaccines. But this is really beyond the scope of this manuscript. As we mentioned above, the main purpose of this manuscript is to provide proof of concept for combining conserved antigens inducing cross-protective antibody response with epitopes activating crossprotective T cell response would against multiple influenza virus infection. Anyhow, we really appreciate your constructive advice. And GC B cells and Tfh cells will be focused in our future study.

Reviewer #2:This paper aims to provide a novel universal influenza vaccine that could induce broad protection against various influenza viruses. For this purpose, the authors designed a new vaccine candidate who is composed of the conserved fragments of nucleoprotein (two epitopes of CTL and helper T lymphocytes), matrix protein 2, highly conserved long α-helix regions of hemagglutinin from group 1 (H1), and group 2 (H3 and H7), and superantigen Staphylococcal Enterotoxin C2, which is named NMHC. They showed that NMHC induces not only antibody production but also T-cell responses. The induced antibodies could bind broad rHA proteins and neutralize H1N1 1A/Michigan/45/2015 (group 1) and H3N2 A/Hong Kong/ 4801/2014 (group 2) viruses.Their proposal is potentially interesting. However, the quality of the results is not sufficient enough to support their conclusion. They concluded that NMHC is a potential candidate for the universal broad-spectrum vaccine for various influenza virus prevention, whereas few data about protection against strains other than H1N1 and H3N2. In addition, there is no comparison to the effectiveness of current vaccines. Neutralization assays not only for H1N1 and H3N2 but also for other strains would strengthen their proposal. If it is difficult to use alive viruses for this purpose, it is recommended to use a simple experimental system such as pseudotyped viruses. For CD4^+^ T cell analysis, various cytokines were induced even in superantigen alone. The authors should show the specificity for T-cell responses against influenza antigens.1) The authors have immunized mice three times every two weeks. The usual current influenza vaccine is immunized twice against mice to test its efficacy. The authors should discuss the comparison of the effectiveness of the current vaccine and theirs.

Thanks for your careful reading and constructive advices. In fact, in our preliminary experiment, we have optimized the dosage and immune frequency of NMHC by detecting the H1N1 and H3N2 specific IgG by ELISA. Based on the result in Author response table 1, we chose a dose of 200 μg each time and each mice for three immunizations in order to achieve a better immune effect. The main objective of this study is to evaluate the feasibility of this recombinant vaccine strategy, and three immunizations for recombinant influenza vaccine have been frequently reported by researchers such as Joan E. M (doi:10.1038/s41541-018-0063-7), Raffael Nachbagauer (doi:10.1038/npjvaccines.2016.15) and John Steel (doi: 10.1128/mBio.0001810). As you mentioned, the usual current influenza vaccine is immunized twice against mice.

Most of the usual current influenza vaccines are inactivated quadrivalent influenza vaccines (QIV) which are more immunogenic than the recombinant vaccine. So two immunizations might be enough for QIV but not for recombinant vaccine. As shown in the supplementary materials (Table S1), the neutralizing antibodies induced by QIV could be effectively detected by HI test but not induced by the recombinant vaccine.

2) They conducted a cytometric bead array to calculate immunoglobulin subclasses in sera presented in Figure 3. In the control group on Day 100, the MFI of IgG2b is reduced compared to other time points. They should calculate the absolute amount of IgG2b, not MFI, by using standard.

Thanks for your careful reading and constructive advices. According to the instruction manual of Mouse Immunoglobulin Isotyping Kit (Cat. No.550026), we used the MFI, the original data, to represent the amount of immunoglobulins. The instruction manual of this kit did not provide how to calculate the absolute amount of each immunoglobulin. Furthermore, MFI was widely used by researchers such as EdwardMorgan (doi:10.1016/j.clim.2003.11.017), Yunxiao Zhao (doi:10.1016/j.cca.2014.09.018) and Yang He (doi:10.1016/j.cca.2012.10.035) to represent the amount of immunoglobulins. As your advice, we tried to calculate the absolute amount of IgG2b by using standards, which contain 0.25μg/ml each of IgG1 κ/λ, IgG2a κ/λ, IgG2b κ/λ, IgG3 κ/λ, IgA κ/λ, IgM κ/λ and IgE κ, and the absolute amount of IgG2b was calculated using the following equation:

MFI IgG2bκ unknown sample / MFI IgG2bκ standard * the dilution multiple *0.25μg/ml

The results showed in the following Author response table 2, the trend is similar to MFI.

3) The data on neutralization assays in Figure 4 is not sufficiently convincing. The authors should perform neutralization assays for H1N1 and H3N2 and other strains to prove that their vaccine candidate is effective against a wide range of influenza viruses. Also, they should conduct a statistical test to show significant differences.

Thanks for your comments and constructive advices. According to your suggestion, we have supplemented neutralization assays for H1N1 A/Brisbane/02/2018 and H3N2 A/Kansas/14/2017 using the same batch of frozen serum samples which were used in neutralization assays for H1N1 A/Michigan/45/2015 and H3N2 A/Hong Kong/ 4801/2014 in our original manuscript. These four strains are all the living influenza virus strains we can obtain in current situation. We are so sorry that we have not been allowed to obtain any other living influenza virus strains to further verify the broad spectrum of NMHC. We agree that neutralization assays for these four strains were not enough to say that NMHC was effective against a wide range of influenza viruses. Therefore, we used the ELISA and Biacore assays, selected the HA fragments of representative influenza virus subtypes, to evaluate the broad binding properties of antibodies induced by NMHC. According to your advice, we reanalyzed the results of neutralization assays and conducted a statistical test to show significant differences by geometric mean titer (GMT). In our revised manuscript, the original Figure 4 has been replaced by the new Figure 4 showed.

4) The ELISA and Biacore assay indicate that the binding capacity of the antibodies induced by vaccination is higher against H2, H5, and H9 than H1, H3, and H7. They should discuss this.

Thanks for your careful reading and comment. As described in Materials and methods part in our manuscript, in the ELISA and Biacore assays, H2, H5, H7, and H9 were commercial recombinant HA proteins, while H1 and H3 were split viruses. It might be that the complex composition and conformation of the split viruses led to the relatively lower binding capacities to H1 and H3 in the ELISA and Biacore results. As to H7, we can see from the Table S2, the ELISA endpoint titers calculated from Figure 5, showed that the titer of H7 induced by NMHC was the same with titers of H2, H5, and H9.

Theoretically, the long α-helix regions of hemagglutinin (amino acids 76–130 from the HA2) were conserved among H1, H2, H3, H5, H7, and H9 (Author response image 1), so the binding capacities of antibodies induced by NMHC to these HAs should be similar. Furthermore, we learned a lot from your advices and we really appreciate your professional comments.

5) The T-cell responses induced by NMHC are not different from those induced by SEC2 alone in Figure 8. The authors should verify whether the induced T cell response is influenza antigen-specific. Specifically, cytokine production should be detected after stimulating splenocytes of post-immunized mice with influenza antigens without SEC2.

Thank you very much for your suggestion. As you mentioned, the T-cell responses in figure 8 was not antigen-specific because that the T cells had not been treated with any antigens in vitro. According to your advice, we have repeated the animal immunization experiment to detect the IL-4 and IFN-γ cytokine production after stimulating splenocytes of post-immunized mice with NMH without SEC2 and SEC2 alone. In brief, female wild-type BALB/c mice were respectively immunized with 200 μg NMHC, 100 μg NMH, 100 μg SEC2, 100 μg NMH plus 100 μg SEC2, and PBS (control) by i.p. for three times with interval of 14 days. Two weeks after the third immunization, splenocytes from each group were collected and treated with 1000 ng/mL SEC2 alone or 1000 ng/mL NMH alone, and PBS was served as CK. Then, the culture supernatants were harvested at 48 h to detect the production of IL-4 and IFNγ by ELISA following the manufacturer’s instructions. In figure 8C, the results showed that, in NMHC, NMH^+^SEC2 and NMH immunized groups, the productions of IL-4 and IFN-γ treated with NMH were significantly higher than CK (*p* < 0.05), while SEC2 and PBS immunized groups didn't show significant changes. This result verified that the T cell response induced by NMH treatment is influenza antigen-specific. However, the splenocytes from all of the immunized groups produced high levels of IL-4 and IFN-γ after SEC2 treatment in vitro, and there were no significant differences among the five immunized groups. As a superantigen, SEC2 could induce nonspecific T cell responses, leading to massive production of cytokine. According to your comment, we have added this supplementary experiment and results to our revised manuscript, and added to Figure 8C in our revised manuscript.

6) Two IgG2b are shown in the Plot representing the standard in Figure 3. I guess that one of the two is IgG2a. Please correct the figure.

Thank you for your careful checks. We are sorry for this typo. In Figure 3A, the second clusters of beads represented the immunoglobulins of IgG2aκ, from top to bottom. And we have revised the WHOLE manuscript carefully to avoid these errors.

7) There is no match between each KD shown in Figure 6 and the KD in the text.

Thank you for your careful checking. We are sorry for this miswriting. We have revised the *Response Unit* charts of Figure 6 C and D. The KDs of H3 and H5 are 685 nM and 72.9 nM, respectively, which is consistent with the text.

8) The sentence "IL-4 (for Th1 subtype), IFN-γ (for Th2 subtype)" should be corrected to "IFN-γ (for Th1 subtype), IL-4 (for Th2 subtype)" (Page 34, Line 558).

Thank you for your careful checks. We have revised this mistyping and check the whole manuscript carefully to avoid such errors.

Reviewer #3:This manuscript describes a vaccine construct comprised of multiple segments derived from influenza proteins, and in some cases conjugated to staphylococcal enterotoxin C2. There is an impressive amount of data and the workup of the immune response is quite detailed, including humoral, cellular, and protection data in mice. Overall, the impression is that this construct has the potential to move forward as a vaccine candidate. In addition, the provided data support their conclusions.The data are clearly presented in most sections, and I am impressed that in places where multiple replicates were performed, the authors listed the number of replicates and the statistical measures.The grammar of the manuscript needs work, and the text would benefit from copy-editing at a minimum and might even benefit from editing for length in some sections.

Thank you for careful checking and constructive advice. We have revised the WHOLE manuscript carefully to avoid language errors. In addition, we have asked the Edanz Group China, a professional English editing agency, to polish our manuscript. We believe that the language is now acceptable for the review process. The following figure shows the certificate of English editing.

In some places, I would probably have presented some of the data differently. For example, I typically display bead-based antibody binding assays on a logarithmic scale which is more in line with the type of data shown. That being said, there is nothing deceptive in how the data are presented, so a change may not be necessary.

Thank you for your suggestion. We think you are referring to the results of Figure 3 in our original manuscript. According to the instruction manual of Mouse Immunoglobulin Isotyping Kit (Cat. No.550026), we used the MFI, the original data, to represent the amount of immunoglobulins. Furthermore, MFI was widely used by researchers such as Edward Morgan (doi:10.1016/j.clim.2003.11.017), Yunxiao Zhao (doi:10.1016/j.cca.2014.09.018) and Yang He (doi:10.1016/j.cca.2012.10.035) to represent the amount of immunoglobulins. Anyhow, we really appreciate your constructive advice.

I do not recommend additional experimentation. I do think that moving this candidate forward through regulatory bodies may face significant hurdles, but that is not the problem being addressed in this paper.

Thank you for your recognition of our work. The main purpose of our study is to provide proof of concept for combining conserved antigens inducing cross-protective antibody response with epitopes activating cross-protective T cell response would against multiple influenza virus infection. The recombinant protein NMHC is rather an example for this strategy than a certain vaccine. The structure of NMHC must be further optimized, and the safety evaluation and effectiveness validation are needed before it become a candidate vaccine. But this is really beyond the scope of this manuscript.

Finally, I think Figure 13 is really more suited to a review article and not appropriate for this manuscript.

Thank you very much for your suggestion. We used Figure 13 to better illustrate the anti-influenza immune response mechanisms induced by NMHC. We can delete it if you and the editor think it is not appropriate for this manuscript.